# An innovative strategy to identify new targets for delivering antibodies to the brain has led to the exploration of the integrin family

**Céline Cegarra**[1¤b]*, **Béatrice Cameron**[2], **Catarina Chaves**[1¤a], **Tarik Dabdoubi**[2¤b], **Tuan-Minh Do**[1¤a], **Bruno Genêt**[3¤b], **Valérie Roudières**[1¤b], **Yi Shi**[4], **Patricia Tchepikoff**[4¤b], **Dominique Lesuisse**[1¤a]

**1** Rare and Neurologic Diseases Research Therapeutic Area, Sanofi, Chilly Mazarin, France, **2** Biological Research, Sanofi, Vitry-Sur-Seine, France, **3** Integrated Drug Discovery, Sanofi, Vitry-Sur-Seine, France, **4** Histology, Translational Sciences, Sanofi, Vitry-Sur-Seine, France

¤a Current address: Sanofi, Chilly Mazarin, France
¤b Current address: Sanofi, Vitry-Sur-Seine, France
* celine.cegarra@sanofi.com

**Data Availability Statement:** All relevant data are within the manuscript and its Supporting Information files.

## Abstract

### Background

Increasing brain exposure of biotherapeutics is key to success in central nervous system disease drug discovery. Accessing the brain parenchyma is especially difficult for large polar molecules such as biotherapeutics and antibodies because of the blood-brain barrier. We investigated a new immunization strategy to identify novel receptors mediating transcytosis across the blood-brain barrier.

### Method

We immunized mice with primary non-human primate brain microvascular endothelial cells to obtain antibodies. These antibodies were screened for their capacity to bind and to be internalized by primary non-human primate brain microvascular endothelial cells and Human Cerebral Microvascular Endothelial Cell clone D3. They were further evaluated for their transcytosis capabilities in three *in vitro* blood-brain barrier models. In parallel, their targets were identified by two different methods and their pattern of binding to human tissue was investigated using immunohistochemistry.

### Results

12 antibodies with unique sequence and internalization capacities were selected amongst more than six hundred. Aside from one antibody targeting Activated Leukocyte Cell Adhesion Molecule and one targeting Striatin3, most of the other antibodies recognized β1 integrin and its heterodimers. The antibody with the best transcytosis capabilities in all blood-brain barrier *in vitro* models and with the best binding capacity was an anti-αnβ1 integrin. In comparison, commercial anti-integrin antibodies performed poorly in transcytosis assays,

**Funding:** The author(s) received no specific funding for this work.

**Competing interests:** All authors are Sanofi employees and may hold shares and/or stock options in the company This does not alter our adherence to PLOS ONE policies on sharing data and materials.

**Abbreviations:** ABC, Antibody Binding Capacity; ALCAM, Activated leukocyte cell adhesion molecule; ARPC5, Actin-related protein 2/3 complex subunit 2; BBB, Blood Brain Barrier; BCRP, Breast Cancer Resistance Protein; BMEC, Brain Microvascular Endothelial Cells; BSA, Bovine Serum Albumin; CD, Cluster of Differentiation; EBM, Endothelial Basal Medium; ELISA, Enzyme-Linked Immunosorbent Assay; ENPL, Endoplasmin; FFPE, Formalin-Fixed Paraffin-Embedded; FPKM, Fragments Per Kilobase per Million mapped fragments; HBSS, Hanks' Balanced Salt Solution; hCMEC/D3, Human Cerebral Microvascular Endothelial Cell; IgG, Immunoglobulin G; IHC, Immunohistochemistry; INSR, Insulin Receptor; IP, Immunoprecipitation; IS, Internalization Score; LAMP1, Lysosomal-Associated Membrane Protein 1; LDL, Low-Density Lipoprotein; NBB, Netherlands Brain Bank; NHP, Non-Human Primate; $P_{app}$, Apparent Permeability; PBS, Phosphate-Buffered Saline; $PC_R$, *Pulse-chase* Ratio; RCN-1, reticulocalbin; RMT, Receptor-Mediated Transcytosis; RSSA, 40S ribosomal protein SA; SM, small intestin; SRSF1, Serine/arginine-rich splicing factor 1; SP16H, FACT complex subunit SPT16; TEER, Trans-Endothelial Electrical Resistance; TFRC, Transferrin Receptor C; ZO, Zona Occludens.

emphasizing the originality of the antibodies derived here. Immunohistochemistry studies showed specific vascular staining on human and non-human primate tissues.

## Conclusions

This transcytotic behavior has not previously been reported for anti-integrin antibodies. Further studies should be undertaken to validate this new mechanism *in vivo* and to evaluate its potential in brain delivery.

## Introduction

The barrier between brain tissues and circulating blood represents a major obstacle in the treatment of central nervous system diseases such as neurodegenerative diseases or brain cancers [1]. The blood-brain barrier (BBB) is only permeable to very small lipophilic compounds [3] and the challenge of crossing it to access the brain parenchyma is especially acute for large polar molecules such as biotherapeutics and antibodies [2, 3]. Following systemic administration, the tissue to blood ratio of antibodies is generally in the range of 10 to 50% [4], whereas for the highly protected brain tissue this ratio is reported in the average of 0.1% [5]. Therefore, increasing brain exposure of biotherapeutics will be key to their success in this field.

So far, the most successful strategy to carry biotherapeutics to the brain has been to use antibodies against receptors such as transferrin [6] and insulin [7] (sometimes referred to as the 'Trojan horse' approach). However, several challenges remain in the field. Specific toxicities can be linked to the modulation of these receptors, as has been shown in the case of Transferrin Receptor C (TFRC) [8–10] and Insulin Receptor (INSR) [7, 11]. These receptors are ubiquitously expressed and therefore effects and liabilities can be spread to several organs and tissues. So far, no brain-specific receptor capable of mediating brain transcytosis has been discovered.

Two main strategies have been used to identify novel mechanisms of brain receptor mediated transcytosis (RMT) applicable to antibodies. The first one has often started with membrane proteins known to be highly enriched in brain microvascular endothelial cells (BMECs). Lipoprotein-related protein (LRP) [12, 13] and Low-Density Lipoprotein (LDL) [14] receptors are respectively able to transport lactoferrin, melanotransferrin, tissue plasminogen activator, β-amyloid precursor protein and LDL, ApoE proteins across the BBB. However, antibodies against lipoprotein-related protein receptor LRP1R did not demonstrate brain exposure enhancement [15], showing that this specific property is not shared by all transmembrane proteins and, to our knowledge, no anti-LDLR antibody has demonstrated enhanced brain exposure. In contrast, Insulin Growth Factor 1 Receptor (IGF1R) [16] has demonstrated this ability to ferry antibodies to the brain. More recently, transcriptomic or proteomic differential analyses have been carried out to identify new brain specific RMT mechanisms. This has yielded proteins such as basigin, glucose transporter 1 (GLUT1) or cluster differentiation (CD)98 [15, 17, 18]; the latter demonstrated efficient delivery of an antibody to the brain.

A second strategy, to identify new brain specific RMT mechanisms, is to screen phage libraries of peptides or antibodies or fragments on a functional assay, either binding to BMECs or transcytosis. The most prominent example of this strategy is the identification of the FC5 and FC44 single domain antibodies from screening a phage display naïve library, where two distinct sequences named FC5 and FC44 were identified [19–23]. FC5 was deorphaned and the target shown to be an α (2,3) sialoglycoprotein [21, 24–27].

Using immune libraries should increase the probability of finding a brain specific target. Even though some precedents can be found in the field of oncology, based on immunization with tumoral cells [28], human epithelial carcinoma cells [29] or glioma cells [30, 31], no application to brain delivery had been reported. At the time we submitted our article, a lamprey immunization with murine brain microvessels plasma membranes with the aim of discovering new brain targeting receptors was published by J.M. Lajoie et al [32]. The first goal of the present report is to describe such an approach of generating an immune library from immunized mice and screening the resulting antibodies based on binding, internalization and finally transcytosis.

Aiming to generate human antibodies against specific brain microvascular membrane proteins, we therefore immunized Trianni Mouse® with fresh primary non-human primate (NHP) BMECs. The resulting antibodies were successively selected for their binding and uptake on BMECs and Human Cerebral Microvascular Endothelial Cell clone D3 (hCMEC/D3), leading to candidates that were further analyzed in three *in vitro* models of transcytosis. Among the handful of antibodies that underwent transcytosis, nine were identified as anti-integrin β1 antibodies. Although peptidic ligands of integrins have been occasionally associated with brain exposure enhancement in the context of nanoparticles, this family of proteins has not been shown to enhance brain exposure of antibodies. Thus, a second objective of the present work has been to investigate the link between the various parameters such as affinity, α and β subunit specificity, functional activity of the monoclonal antibodies and transcytosis in order to select the best candidate for *in vivo* validation.

## Materials and methods

### Proteins, reagents, cell lines

Recombinant integrin proteins were purchased from OriGene for human monomer α3, α5 and β1 (tp320975, tp301151 and tp303818), from GeneTex for human monomer α4 (GTX48181), from R&D Systems for human and mouse dimer α3β1, α4β1 and α5β1 (2840-a3, 3230-a5, 5668-a4, 7728-a5, 9374-a3) for human ALCAM 656-AL, from Sino Biological for human α5β1 (CT-014-H2508H), from Abcam for Striatin3 (ab162295)

Antibodies were purchased from antibodies-online (natalizumab ABIN5668196), from Abcam (Anti-VE Cadherin ab33168), from BD Biosciences (553715), from BioLegend (343802 and MFR5 103801), from Interchim (DCABH-8217), from Invitrogen (14-0299-82, MA5-17103, MA1-25298), from Millipore (MABT409, MABT199, MAB2079Z) from Novus Biological (NBP2-52708), from Proteintech (66070-1-Ig), from R&DSystems (MAB1345), from Sigma-Aldrich (MAB1965), from ThermoFisher Scientific (Anti-ZO-1 #61–7300; Anti-Occludin #33–1500; all other antibodies were produced in-house by Sanofi Biological Research.

hCMEC/D3 cells were obtained from Cedarlane. Cells were cultured in the "Cell biologics" medium supplemented with H1168. PC3 were obtained from DSMZ (ACC465). Cells were cultured in 45% Ham's F12 + 45% RPMI 1640 + 10% FBS. These human cells are growing adherently in monolayers.

### Animals

Experiments were performed at Sanofi in our Association for Assessment and Accreditation of Laboratory Animal Care (AAALAC)-accredited facilities in full compliance with standards for the care and use of laboratory animals, according to the French and European Community (Directive 2010/63/EU) legislation. All procedures were approved by the local animal ethics committees (Ethical Committee on Animal Experimentation (CEEA) #24 and #21), of Sanofi,

Vitry-Alfortville and Chilly Mazarin Research Centers, France, and the French Ministry for Research.

Male and female *Macaca fascicularis* non-human primates (NHPs) were purchased from Le Tamarinier and Noveprim Ltd. (Mahebourg, Mauritius). They were aged from 4.8 to 5.9 years. Animals were group-housed in aviaries or interconnected mobile cages. They were housed under controlled conditions (20–24˚C, 40–70% humidity, 10–15 renewals per hour of filtered, non-recycled air, 12-h light cycle) with free access to filtered tap water and daily distribution of expanded diet and fruits or vegetables. Animals from which brain microvessels were harvested had previously been used in pre-clinical studies; they were submitted to a drug washout period of at least 1 month before euthanasia and brain collection. Animals were deeply anesthetized with Zoletil 5O (0.2 ml/kg IM) followed by administration of pentobarbital (0.15 ml/lg IV). Animals were sacrificed by exsanguination then brain was collected.

Pregnant Sprague-Dawley rats were purchased from Janvier Labs (France) between E10 and E12. Upon arrival, rats were housed individually in an enriched environment in a pathogen-free facility at a constant temperature of $22 \pm 2$˚C and humidity ($50 \pm 10$%) on a 12-h light/dark cycle with ad libitum access to food and water. Animals were anesthetized with isoflurane and sacrificed by guillotine then brain was collected.

Immunizations of TRIANNI Mouse® (San Francisco, CA, USA) were performed at Sanofi (Vitry-sur-Seine, France) and overseen by a licensed veterinarian. Institutional Animal Care and Use approval was obtained by the CEPAL committee (procedure #PEA 2012–0077). TRIANNI Mouse® were housed in an enriched environment in a pathogen-free facility at a constant temperature of $22 \pm 2$˚C and humidity ($50 \pm 10$%) on a 12-h light/dark cycle with ad libitum access to food and water.

## Primary cell production

**Isolation of brain microvessels from NHP cortex.** Brains from NHPs were collected shortly after euthanasia in ice-cold Hibernate A medium (Thermo Fisher Scientific). All subsequent steps were performed at 4˚C and under a biological safety cabinet. Brain cortex was isolated and placed in petri dishes containing ice-cold Hanks' Balanced Salt Solution (HBSS). The meninges and the cortical white matter were removed. The collected tissues were transferred into a new sterile container with HBSS, finely minced with a scalpel, and then pelleted by centrifugation (5 min at 600 g, 4˚C). The pellet was resuspended in a collagenase/dispase® solution (Roche, Meylan, France, Collagenase 0.1 U/mL; Dispase 0.8 U/mL prepared in $Ca^{2+}/Mg^{2+}$ free HBSS) containing type I DNAse at 20 U/mL and TLCK at 0.147 µg/mL, and incubated at 37˚C for 60 min, under vigorous agitation. The digested tissue was carefully homogenized, and centrifuged for 5 min at 600 g, 4˚C. The resultant pellet was resuspended in HBSS containing 20% Bovine Serum Albumin (BSA) and centrifuged for 30 min at 2000 g, 4˚C. The myelin ring-containing supernatants were discarded, and the vessel-containing pellet was resuspended and re-incubated in the collagenase/dispase® solution in presence of DNAse and TLCK for 30 min at 37˚C. This suspension was re-pelleted by centrifugation (5 min at 600 g, 4˚C), and the final pellet (named P0D0 fraction from this point onwards) was resuspended in endothelial basal medium (EBM)-2 supplemented with Kit EGM-2 MV SingleQuots (Lonza, Basel, Switzerland) containing 3 µg/mL puromycin, seeded in pre-coated (collagen IV 100 µg/mL, fibronectin 10 µg/mL, Sigma, Saint Quentin Fallavier, France) cell culture flasks, and incubated at 37˚C under 5% $CO_2$ for 7 days. Every two days the cell medium was changed and the supplemented puromycin concentration lowered to 2 µg/mL, and subsequently removed. Following seven days of expansion, at P0D7, BMECs from cortex were further singularized and re-plated de novo for an additional seven-day cell expansion (P1D7).

**Isolation of brain microvessels from human brain.** All human brain samples were obtained from The Netherlands Brain Bank (NBB), Netherlands Institute for Neuroscience, Amsterdam (open access: www.brainbank.nl). All Material has been collected from donors from whom a written informed consent for brain autopsy and the use of the material and clinical information for research purposes had been obtained by the NBB. Human brain samples were preserved in Hibernate A (Gibco™ A1247501) medium during transport à 4˚C and microvessels were isolated as described above for NHP brain samples. All experiments were conducted in a biosafety level 2 laboratory by trained personnel.

**Cell preparation for Brainplotting™ models.** Human tissue was provided by Brainplotting™ (iPEPS, Institut du Cerveau et de la Moelle épinière, Hôpital Universitaire de la Pitié-Salpêtrière, Paris, France) in partnership with Sainte-Anne Hospital, Paris (neurosurgeon Dr. Johan Pallud) and harvested during scheduled tumor resection surgeries with written informed consent from the patients (authorization number CODECOH DC-2014-2229). Human brain microvessels were obtained from surgical resections of several patients with diffuse oligoastrocytic glioma. Microvessels were isolated from healthy peritumoral brain tissue using an enzymatic procedure [33] adapting methods previously published for rats [34, 35]. Briefly, tissue samples were carefully cleaned from meninges and excess of blood; then, an enzymatic mix was used to dissociate the tissues and microvessels were isolated by retention on a 10 μm mesh. Cells were cultured in EBM-2 medium (Lonza, Basel, Switzerland) supplemented with 20% serum and growth factors (Sigma) [34, 35]. After seeding brain capillaries in petri dishes, primary BMECs were amplified and seeded (P1) on Transwell® (Corning) with microporous membranes (pore size: 0.4 μm) in monoculture. After transcytosis assays, cells could be fixed with ethanol for immunostaining.

**Cell preparation for PharmaCo-Cell® BBB model.** Cells were prepared following the instructions of the ready-to-use MBT24h NHP BBB kit (PharmaCo-Cell®). Briefly, the triculture BBB model with NHP BMECs, pericytes and astrocytes was thawed in warmed medium provided in the kit. After 1h and then 24h the medium was changed. Trans-endothelial electrical resistance (TEER) was measured 4 days after thawing and transcytosis assays performed 1 day later.

**Co-culture experiments.** NHP BMECs were seeded onto 12-well plate Transwell® (Corning Transwell® 3401 polycarbonate filters, 0.4 μm pore diameter; 1.12 cm$^2$, Corning, Sigma), previously coated with collagen IV/fibronectin (Sigma) at a density of 60,000 cells/well in supplemented EBM-2 medium. Immediately following plating, BMEC-coated Transwell® were then placed into plates with either no other cell type (monoculture), or with rat primary astrocytes (80,000 cells/well) plated onto the bottom of the plate 4 days before, in MEMα/F12 cell medium (1:1), 10% FBS, 1% PSN and 5 ng/mL βFGF. Astrocytes were isolated from brains of E18 rat embryos then frozen after one week in culture. All transcytosis experiments were conducted following 4 days or 7 days and in monoculture or co-culture with astrocytes.

**Resistance measurements.** TEER was measured following the culture of BMECs onto Transwell® filters. Resistance was recorded using an EVOM2 epithelial voltohmmeter coupled to a cell culture cup chamber (ENDOHM-12G) (World Precision Instruments, Hitchin, Hertfordshire, United Kingdom). TEER values are presented as Ω. cm$^2$ following subtraction of an un-seeded Transwell® and multiplication by 1.12 cm$^2$ to account for the surface area [33]. TEER measurements were taken three independent times on each sample and at least on triplicate filters for each experimental condition.

**Immunofluorescence.** Cultured cells were fixed in 4% paraformaldehyde for 15 min, at RT, and subsequently permeabilized and blocked in Odyssey LiCor Blocking Buffer containing 0.2% Triton X-100. Primary antibodies were incubated overnight at 4˚C and appropriate secondary antibodies conjugated with Alexa fluorophores (Invitrogen) and Hoechst 33432

(Invitrogen) for nuclei staining were subsequently used for 2h at RT. Images were acquired on a Perkin Elmer Operetta CLS system.

**RNA sequencing.** As described in Chaves et al [33], the RNAseq libraries were prepared with 30 ng of input total RNA using the NEB Next Ultra II Directional RNA Library Prep Kit for Illumina (New England Biolabs, #E-7765S) with the NEB Next rRNA Depletion Kit (New England Biolabs, #E6310L) and following the manufacturer's instructions. The libraries were then paired end sequenced (75 cyclesx2) on the NovaSeq 6000 instrument (Illumina) using the NovaSeq 6000 SP Reagent Kit (300 cycles; #20027465, Illumina). RNA-seq data analysis was performed using ArrayStudio (Qiagen). Briefly, raw data QC is performed then a filtering step is applied to remove reads corresponding to rRNAs as well as reads having low quality score or shorter than 25 nt. Reads were further mapped to the Cyno Washington University 2013 genome, based on the contigs assembled from a WGS project submitted by Washington University in 2013, using OSA4 (Omicsoft Sequence Aligner, version 4, Qiagen) and quantified using Ensembl. R94 model of transcriptome, paired reads were counted at gene level. Differential analysis of gene expression was performed at gene level using DESeq2. The variable multiplicity was taken into account and false discovery rate adjusted p-values calculated with the Benjamini-Hochberg correction. Functional analysis of differentially expressed genes was performed using IPA (Qiagen). Gene Set Enrichment analysis was performed using Ingenuity and GSEA software. High-throughput sequence data are available on the Gene Expression Omnibus under the GSE154901 accession number. Read depth from RNAseq was counted as fragments per kilobase per million mapped fragments (FPKM). FPKM read counts were obtained for each BEC fraction (P0D0, P0D7). A minimum of 1 FKPM in at least half of the observations per group, and in at least one of the groups, was required for a gene to be considered as expressed and included in the analysis.

## Antibody generation

**Mouse immunization and hybridoma selection.** The Trianni Mouse® is a C57BL/6 strain that transgenically expresses a complete repertoire of fully human immunoglobulin G (IgG) and immunoglobulin kappa (IgK) V(D)J genes, but retains mouse regulatory genomic sequences [36]. The Repetitive Immunization at Multiple Sites (RIMMS) method was used as described by Kilpatrick et al. [37]. In this approach, 6-8-week-old female mice each received four rounds of subcutaneous injections of $10^6$ NHP primary BMECs per animal over a course of 14 days at intervals of 3–4 days. Cells emulsified in Titermax's adjuvant (TiterMax® Gold Adjuvant; Sigma #T2684) were administered subcutaneously to six sites proximal to draining lymph nodes, along the back of the mice and to six juxtaposed sites along the abdomen. Four days after the last injection, mice were sacrificed. Bilateral popliteal, superficial inguinal, axillary, and branchial lymph nodes were isolated for antibody generation. B cells were isolated from the lymph.

Single-cell suspension was fused with P3X63-AG8.653 myeloma cells using the polyethylene glycol fusion method [38]. After incubation at 37˚C for 16–24 hours, the resulting cell preparation was transferred into selective semi-solid medium, plated out into Petri plates and incubated at 37˚C. Ten days after initiation of selection, isolated hybridoma colonies were picked and amplified using ClonePix™ 2 Mammalian Colony Picker.

**VH/VL sequence retrieval and monoclonal antibody production.** Paired VH/VL genes were retrieved from 100 clonal cells by RT-PCR and sequenced using a protocol similar to the one described in Tiller et al. [39].

Nucleic acid sequences coding for the antibody heavy or light chains were cloned into mammalian expression plasmids under the CMV enhancer/promoter and the SV40 polyA

signal. Resulting plasmids were transfected into Human Embryonic Kidney (HEK) 293 cells (Thermo Fisher Scientific; K9000-10) using FreeStyle™ MAX 293 Expression System according to the manufacturer's instructions. Monoclonal antibodies were produced at 30 mL scale, purified by protein A affinity chromatography and stored in Phosphate-Buffered Saline (PBS) after desalting on mini trap Sephadex G-25 column.

## Antibody characterization by flow cytometry and internalization

**Flow cytometry.** The apparent affinity $EC_{50}$ of the antibodies to NHP BMEC primary cells and hCMEC/D3 cell line was measured by flow cytometry. Cells were plated at 100 000 cells/well on 96-well plates (Falcon; 353910) and 100 μL/well of antibody was added in 2-fold serial dilutions starting at 300 μg/mL up to 12 dilutions in assay diluent for 20 min at 4˚C and washed twice in PBS with 1% BSA. Binding of the antibodies was detected by 100 μL/well of Alexa Fluor® 488 conjugated goat anti-human IgG (Jackson Immunoresearch; #109-545-098) for 15 min at 4˚C and then washed twice in PBS with 1% BSA. The antibody apparent affinity $EC_{50}$, is the half maximal effective concentration representing the apparent affinity of the antibody to its target. $EC_{50}$ was evaluated after centrifugation and resuspension of cells by adding 150 μL/well of PBS with 1% BSA and read using Guava® easyCyte 8HT Flow Cytometry 5 System. $EC_{50}$ values were estimated using the 4-parameter logistic model according to Ratkowsky and Reedy [40]. The adjustment was obtained by nonlinear regression using the Levenberg-Marquardt algorithm in SAS© software.

**Antibody binding capacity (ABC).** Levels of receptor density were determined by flow cytometry using Biocytex Human IgG Calibrator kit (# CP010) in hCMEC/D3. Briefly, a primary antibody was used at 10 μg/ml for detecting surface receptors on each cell line using the manufacturer's protocol. Surface densities were calculated using Biocytex kit calibration standards and formulation provided by the kit.

**Internalization.** An ImageStreamX Mark II multispectral imaging flow cytometer (Luminex Corp.) was used to monitor the internalization of the antibodies following binding to hCMEC/D3 cell line and NHP BMECs. Viable cells ($4 \times 10^4$ cells) were seeded into wells of 6-well plates and incubated with 5 μg/mL of monoclonal antibodies for 18 hours at 37˚C under 5% $CO_2$ or 1 hour at 4˚C in parallel. Cells were washed by centrifugation in PBS with 1% BSA at 400 g for 3 minutes. Cells were fixed and permeabilized using 500 μL of BD Cytofix/Cytoperm buffer (BD Biosciences; 554722) on ice for 20 minutes. Cells were washed by centrifugation in 4 mL of Perm/Wash Cell buffer (BD Biosciences; 554723) at 400 g for 3 minutes. To test whether antibodies could be internalized, 1 mL of a 1:400 dilution of Alexa-Fluor488 conjugated goat anti-human IgG (Jackson Immunoresearch; 115-545-164, West Grove, Pa.) was incubated on ice for 20 minutes. After incubation, 4 mL of PBS with 1% BSA buffer was added to wash, before centrifuging (400 g, 3 min). The supernatant was flicked from the plate before the cells were fixed with 150 μL 1% formaldehyde on ice for 20 minutes. The fluorescence of cells was analyzed with the ImageStream multispectral imaging flow cytometer using the Internalization feature. Five thousand events were acquired for each experimental condition and the corresponding images were analyzed using the IDEAS 5 image-analysis software. The internalization score (IS) was then computed as previously described [41] by using the equation below:

$$IS = \log\left(\frac{a}{1-a}\right) \text{ where } a = \left(\frac{mI}{mI + mB}\right) * \left(\frac{pI}{PB}\right)$$

B = External and I = Internal part of the cells, mI = Mean intensity of upper quartile pixels in I, mB = Mean intensity of upper quartile pixels in B, pI = Peak intensity of upper quartile pixels in I, pB = Peak intensity of upper quartile pixels in B.

During the early screening, the normalized IS was plotted. It was the calculated ratio of the IS measured at 37˚C divided by the IS at 4˚C.

## Transport assays

***Pulse-chase* assays.** *Pulse-chase* transport assays were performed with non-contiguous cells like hCMEC/D3 or NHP BMECS with TEER measurement below 150 $\Omega.cm^2$.

The first step was a pulse where test target antibodies (1 μg/mL, anti-integrin or reference anti-human/NHP TFRC) were added to the upper chamber on day 4. Fresh EBM medium without antibodies was added to the bottom chamber. After 2h at 37˚C, 3 washes with PBS were performed. The second step was a chase at 37˚C with fresh EBM medium added to the top and bottom chambers. Final aliquots from both chambers were taken 4h following incubation at 37˚C under 5% $CO_2$. Transwell® membranes were washed 3 times with PBS, removed with a scalpel and frozen with 200μL of water to lyse cells after 3 freeze/thaw cycles. Antibody levels in cells, luminal and abluminal compartments were determined by Enzyme-Linked Immunosorbent Assay (ELISA) (MESOQuickPlex SQ120, MesoScale Discovery, Rockville, MD, USA). Relative transport classification was done by calculating the pulse /chase ratio $PC_R$:

$$PCR = \frac{Qbasolateral}{Qtotal}$$

Where $Q_{basolateral}$ = antibody quantity in ng measured in the bottom chamber, $Q_{total}$ = antibody quantity in ng measured in cells and top and bottom chamber.

Outliers were defined by $Q_{total} < 10\%$ median of $Q_{total}$ samples and were removed from analysis

For each parameter, only antibodies with at least 3 measurements were included in statistical analysis. For pulse chase ratio analysis, as no negative control was available, a superiority analysis was conducted for each antibody using a one sample one-sided t-test analysis versus constant. The superiority margin was defined as 0.1 for pulse chase ratios. In order to control multiplicity at 2.5% level, the p-values were adjusted using the Bonferroni-Holm correction. The analysis was performed using SAS 9.4 for Windows 10.

**Transcytosis assays.** Test antibodies (1 μg/mL, anti-integrin or reference anti-human/ NHP TFRC) or human holo-transferrin (2 μg/mL) and control antibodies without target on cells (1μg/mL, mouse IgG, clone MG1-45, BioLegend or human IgG anti-TNP) or FITC-coupled 70 KDa Dextran (10 μg/mL) were added to the upper chamber on cultures with TEER values from 150 $\Omega.cm^2$. Fresh EBM medium without antibodies was added to the bottom chamber. Final aliquots from both chambers were taken 240 min following incubation at 37˚C under 5% $CO_2$. Compound levels in stock solutions (t = 0min), upper and lower compartments (t = 240min) were determined by ELISA (MESOQuickPlex SQ120, MesoScale Discovery). Apparent permeability coefficients ($P_{app}$) in $cm.min^{-1}$ were calculated using the following formula:

$$Papp = \frac{V \times Cabluminal}{A \times Cluminal \times t}$$

where V = volume of cell medium in the bottom chamber (mL), A = surface area of the insert ($cm^2$), $C_{luminal}$ = compound concentration loaded in the upper chamber (μM), $C_{abluminal}$ = compound concentration measured in the bottom chamber (μM); t = time of the assay (min).

Outliers were defined by $P_{app}$ IgG control > 50% median of $P_{app}$ of all IgG control and were removed from analysis.

For each parameter, only antibodies with at least 3 measurements were included in statistical analysis. For Papp ratio, a superiority analysis was conducted using an analysis of variance (ANOVA) to compare each antibody versus negative control followed by one-sided Dunnett's test to control multiplicity at 2.5% level. Prior to statistical analysis, data were subjected to inverse transformation to ensure normality of residuals. The analysis was performed using SAS 9.4 for Windows 10.

**ELISA method.** Standard 96-well SECTOR plates (Meso Scale Discovery) were coated with 0.5 μg/mL of human Fab'2 or mouse IgG or with 0,75 μg/mL of recombinant integrin protein in PBS and then incubated for 1 h under agitation at room temperature. After incubation, plates were washed three times with PBS-Tween 0.05% (Calbiochem, 524653) and blocked for 1 h at room temperature with 0.1% BSA solution (A7030, Sigma). After blocking the plates, antibody serial dilutions (from 10 to $10^{-6}$ μg/mL) or samples collected in transport assays and standards were incubated on plates for 2 h at room temperature. After incubation, plates were washed three times with PBS-Tween 0.05%, and bound antibody was detected with SULFO-TAG conjugated goat anti-mouse antibody (R32AC-1, Meso Scale Discovery) using TPA containing read buffer (R92TC-2, Meso Scale Discovery). Concentrations were determined from the standard curve using a four-parameter non-linear regression program (Discovery Workbench version 4.0 software). $EC_{50}$ values were estimated using the 4-parameter logistic model according to Ratkowsky and Reedy [40]. The adjustment was obtained by non-linear regression using the Levenberg-Marquardt algorithm in SAS© software.

## Antibody target identification

**Target identification by immunoprecipitation and LC-MS/MS peptide mapping.** Each monoclonal antibody target was purified from a hCMEC/D3 membrane fraction under mild denaturing conditions using Pierce Classic immunoprecipitation Kit (#26146) according to the manufacturer's instructions. Pulled-down proteins were separated on SDS-PAGE stained with Coomassie blue. Stained protein bands were submitted to an in-gel tryptic digestion by Digestpro MS (Intavis) after reduction in 10 mM DTT and alkylation with 55 mM iodoacetamide. Eluted peptides were analyzed by tandem mass spectrometry (LC-MS/MS) on a Q-Exactive Plus benchtop mass spectrometer (Thermo Fisher Scientific). Peptides were searched in database with MaxQuant/Perseus® against UniProt Human. The following filters were carefully applied to obtain the list of best target candidates: at least two unique peptides, filter on known or observed contaminants with a negative control. Identified soluble proteins were purchased from vendors and used by ELISA to confirm binding to the monoclonal antibody. Human recombinant proteins used were: Activated Leukocyte Cell Adhesion Molecule (ALCAM)/ CD166: R&D System 656-AL; α5β1: Sino Biological CT-014-H2508H, α3β1: R&D Systems 2840-A3, Striatin3: Abcam ab162295.

**Target identification by Retrogenix™/Charles River cell microarray technology.** Each monoclonal antibody was provided to Retrogenix™/Charles River for assessment using their human cell microarray technology. Expression vectors encoding over 5,205 full-length human plasma membrane proteins were spotted onto microarray slides. Human cells grown over the top become reverse-transfected resulting in cell surface expression of each respective protein at distinct slide locations. The antibody was applied, and specific binding analyzed and confirmed using an appropriate detection system.

In addition, identified anti-integrin and control antibodies were added to slides of fixed HEK293 cells overexpressing a series of 22 naturally occurring integrin heterodimers and

slides of fixed untransfected HEK293 cells as controls. Specific binding was then analyzed as previously.

## Immunohistochemistry (IHC)

**Chromogenic IHC.** Post-mortem human brain samples for IHC were obtained from external biological resource centers in full accordance with legislation and ethical standards. TFRC antibody was evaluated on PC3 and hCMEC/D3 cells to evaluate formalin-fixed paraffin-embedded (FFPE) and frozen sample conditions. The result was confirmed on FFPE and frozen hCMEC/D3 cells to determine study feasibility for the test antibodies. The frozen samples were chosen. Four human frozen cortex sections and five NHP frozen (2 cortex and 3 hippocampal) sections were from an internal collection. Samples were assessed with a reference TFRC antibody. Internal integrin antibodies were firstly evaluated on PC3 cells, secondly on NHP brains then on other NHP tissues (brain, pancreas, kidney, lung, liver, heart, arterial and cardiac muscles, meninges, basement membrane). The IHC pattern in NHP was compared to human (brain, heart, kidney, pancreas and liver, artery confirmed by anti-SMA, vein and capillaries). IHC staining was performed using the Ventana Discovery XT automated System (Ventana Medical Systems, Inc., USA) with a standard streptavidin-biotin-labeling technique. All detection systems were manufactured by Ventana Medical System Inc. IHC optimal concentrations for each antibody were determined by individual pilot studies under the standard condition. Sections derived from FFPE samples were pretreated with antigen retrieval procedure (Heat Induced Epitope Retrieval) and tris borate EDTA buffer (pH8-8.5). After endogenous biotin blocking for 4 minutes for avidin/biotin steps, the primary antibody was incubated for 60 minutes at 24˚C, with a concentration of 0.5 μg/mL of TFRC antibody or 1 μg/ml of 4F2. As a negative control, a mouse IgG1 isotype was used instead of the primary antibody. The secondary antibody, a goat anti-mouse biotin-conjugated IgG1 antibody (1070–08, Southern Biotech, USA), was incubated at 24˚C for 32 minutes at a final dilution of 1/200.

Immunostaining was done with DABMap™ chromogenic detection kit according to the manufacturer's recommendations (760–124, Ventana). A counterstaining step was done with hematoxylin (760–2021, Ventana) and bluing reagent was applied (760–2037, Ventana). Stained slides were dehydrated, and cover slipped with cytoseal XYL (8312–4, Richard-Allan Scientific, USA).

Immunostaining sections were observed with a Nikon Eclipse E400 bright-light microscope and slides were scanned and digitized using the ScanScope XT system (Aperio Technologies, Vista, CA). Digitized images were then captured using the ImageScope software (version 9.1, Aperio Technologies) at x20 magnification.

**Fluorescent IHC.** Five frozen NHP cortices were from an internal collection. IHC was performed using the Ventana Discovery Ultra from ROCHE with a standard protocol. All detection systems were manufactured by Ventana Medical Systems Inc. IHC optimal concentrations for each antibody were determined by individual pilot studies under the standard condition. Primary antibodies (1 μg/mL anti-CD31 antibody or 0.2 μg/ml 4F2) were incubated on tissue sections for 1 h. The secondary antibody anti-mouse conjugated with Cy5 for TFRC and with FITC for 4F2 incubated for 8 min and then 8 min with Dapi (D1306, ThermoFisher Scientific). Sections were imaged with an Olympus VS120 scanner at 100X.

## Results

### Preparation of human primary BMECs

We needed to have a significant amount (around 200 million for complete immunization or 20 million using the RIMMS method) of primary human BMECs for our immunization

experiments. Primary BMECs can be isolated from fresh brain tissue, either from deceased patients [42–44] or from patients having undergone surgical resections of epileptic areas or glioblastomas [17, 45–51]. Because the latter option can present a risk of accessing tissue potentially contaminated with damaged or too permeable tissue, we focused on obtaining post-mortem brains. Through a collaboration with the Netherlands Brain Biobank, in which we established a series of exclusion criteria based on age and pathological conditions, we were able to obtain three samples (approximately 100g each) of fresh human brains within a post-mortem interval of less than 24 hours. Human BMECs from NBB tissues were isolated, cultured and amplified. To obtain the purest possible endothelial cell cultures, we cultured the cells for 7 days in the presence of puromycin to eliminate all cells that did not express P-gp efflux pump. A sample was taken and then characterized during the first passage (P1D0), 3 days after the first passage (P1D3) then 4 days after the second passage (P2D4). S1A Fig shows these cultures which appeared to exhibit the expected phenotype.

We characterized the primary BMECs using qPCR by quantifying specific genes for different cell types likely to be present in the cultures (S1B Fig). We amplified CD31, Claudin5 (CLDN5) (tight junctions) and Breast Cancer Resistance Protein (BCRP) (efflux pump) as endothelial cell markers, GFAP, IBA1 and PDGFRβ as markers of astrocytes, microglia and pericytes, respectively. The results in S1B Fig show that even if the cells exhibit a brain endothelial gene profile upon first isolation (P1D0) with high expression of CD31, CLDN 5 and BCRP and no expression of genes from astrocytes, microglia or pericytes, further culture slowly enriches the cells in pericytes at the expense of endothelial cells as shown by the growing amount of PDGFRβ and the lower amounts of CD31 over time. Thus, this paradigm could not yield the quantities of primary BMECs needed to conduct an immunization campaign.

We therefore turned to NHP BMECs since we could obtain fresh brains with very short post-mortem intervals from animals that were to be euthanized in our in-house *Macaca Fascicularis* animal facilities.

## Preparation and validation of primary NHP BMECs for immunization

A pool of 20 million NHP BMECs were produced as described in Chaves et al [33] characterized and stored frozen at -150˚C until use. We constituted a cell pool from several NHPs to increase genetic diversity in NHP BMECs. NHP BMECs characterization is shown in Fig 1.

Endothelial phenotype is observed by immunofluorescence in Fig 1A. The endothelial phenotype was confirmed with RNA-Seq analysis during the cell culture in Fig 1B. The BBB in vitro transcytosis model functionality is verified in Fig 1C. $P_{app}$ transferrin was compared to $P_{app}$ dextran 70kDa. Transferrin transport was significantly higher than dextran 70 kDa which was not transported through endothelial cells. This transcytosis model is functional, so produced NHP BMECs are functionnal.

## Generation and selection of antibodies binding to NHP BMECs

TRIANNI Mouse® were immunized with the previously validated primary NHP BMECs as immunogen using the Repetitive Immunization at Multiple Sites (RIMMS) approach. Monoclonal antibodies were isolated from the hybridoma technology, and 636 IgGs were established. Screening results are presented in Fig 1D.

Upon screening by flow cytometry, 62 monoclonal antibodies were selected for binding on NHP primary BMECs with or without binding to hCMEC/D3 cells and 31 were found to internalize into NHP primary BMECs or hCMEC/D3 cells at 37˚C compared to 4˚C.

Normalized IS should be interpreted as follows: IS at 37˚C / IS at 4˚C.

IS ≤ 0.2 no internalization

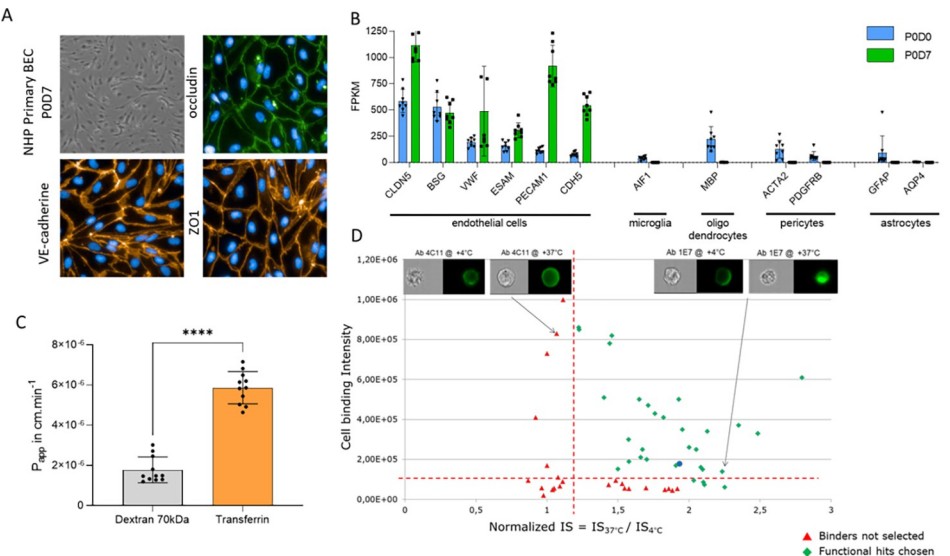

**Fig 1. Cell production and antibody screening.** (A) NHP primary brain cells isolated at passage 0 after 7 days in culture. Cells were observed 100X magnification with inverted microscope and 200X magnification with fluorescent microscope. Immunofluorescence staining: ZO1 and VE-cadherin labelling in orange, occluding in green and nuclei with Hoechst in blue (B) NHP primary BMECs characterized by RNA Seq. Endothelial, microglial, oligodendrocytes, pericytes and astrocytes markers are represented. Blue bars represent analysis from microvessels P0D0 and green bars represent analysis from P0D7 cultured cells (C) Transferrin and Dextran 70kDa transcytosis in NHP BMECs. Results are given as the $P_{app}$ in cm.min$^{-1}$ The statistical significance of differences between groups was analyzed using GraphPad Prism v9.0.0 software (GraphPad Software) and Mann–Whitney test p-value< 0.0001. (D) Monoclonal antibody selection into NHP BMECs. The two selection criteria were cell binding and normalized IS. Green diamonds were selected, red triangles were rejected. The blue point was the anti-TFRC antibody added as a control. Pictures show 4C11 as not internalized antibody and 1E7 as internalized antibody.

$0.2 \leq IS \leq 0.5$ low internalization
$0.5 \leq IS \leq 1$ moderate internalization
$1 \leq IS \leq 2$ high internalization

Variable heavy and light chain sequences of these 31 antibodies were retrieved, 22 antibodies had unique sequences and were further produced and analyzed. From these 22 antibodies, 12 were cross-reactive with hCMEC/D3 cells. $EC_{50}$ values on hCMEC/D3 cells were determined (Table 1). The normalized IS determined showed that all 12 antibodies were able to be internalized (Table 1). ABC is the number of monoclonal antibodies a sample will bind, and correlates to the number of antigens expressed on the cell surface. It was determined by measuring the binding by flow cytometry. The value was correlated to the receptor binding density using the calibration curve as described in Materials and Methods. ABC showed values ranging from 3 800 to more than 173000, suggesting that different epitopes or targets could be recognized at the cell surface of hCMEC/D3 cells (Table 1).

4F2 antibody had the highest antibody binding capacity. These 12 antibodies were further evaluated for their ability to perform transcytosis in three different models.

## Evaluation in three transcytosis models

From Table 1, the 12 monoclonal antibodies displaying binding to both hCMEC/D3 and NHP BMECs were selected, along with a negative control that did not bind to human hCMEC/D3 cells (8C12) to engage in "*pulse-chase*" assays on human hCMEC/D3 and primary NHP BMECs, and in a NHP transcytosis model. We also included an anti-hTFRC antibody for

**Table 1. hCMEC/D3 binding of 12 purified monoclonal antibodies, IS and ABC.**

| clone | $EC_{50}$ nM | Normalized IS | $10^3$ ABC |
|---|---|---|---|
| 1E7 | 1.0 | 2.3 | 72 |
| 3B8 | >540 | 1.7 | 7.8 |
| 3C5 | 1.6 | 1.5 | 24.5 |
| 3E8 | >540 | 2 | 7.8 |
| 4D2 | 5.2 | 1.2 | 61 |
| 4F2 | 1.9 | 1.4 | >173 |
| 6A10 | >540 | 2 | 3,8 |
| 6C7 | 22.5 | 1.2 | 164 |
| 6D6 | >540 | 1.7 | 14.5 |
| 6F5 | 0.6 | 1.8 | 41.5 |
| 8C10 | 0.6 | 2 | 35 |
| 9F4 | 0.8 | 2.1 | 22.3 |
| anti-TFRC | Not determined | 1.9 | Not determined |

which transcytosis had already been demonstrated [33]. All following data for evaluation in the three transcytosis models were reported in S1 and S2 Tables.

**hCMEC/D3 *pulse-chase* experiments.** *"Pulse-chase"* experiments were conducted as described by Sade et al. [52] to evaluate transcytosis capabilities in cells without strong tight junctions (Fig 2A). We used the same protocol for hCMEC/D3 cells and for NHP BMECs for which we had determined a TEER value below 150 $\Omega.cm^2$.

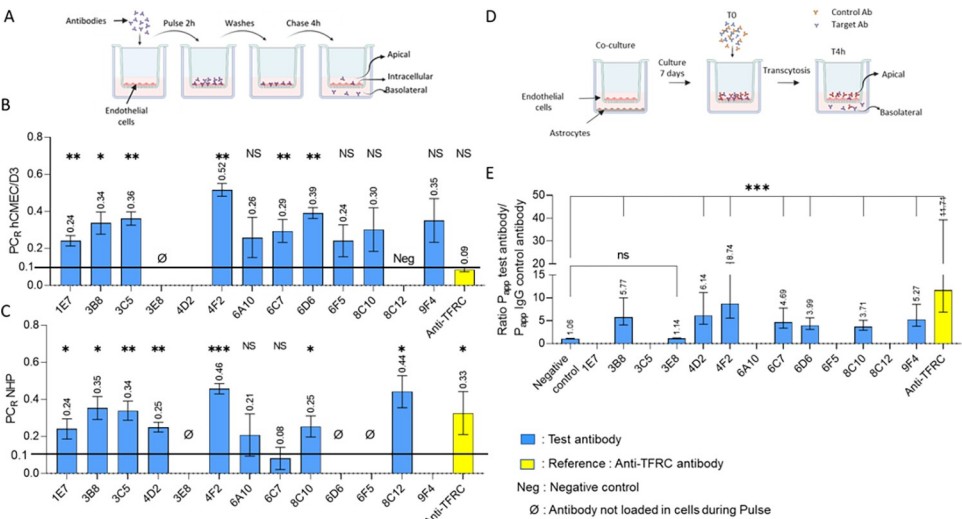

**Fig 2. Antibody transport in three *in vitro* BBB models.** (A) Diagram of a *Pulse-chase* experiment. (B and C) *"Pulse-chase"* ratios ($PC_R$) determined in hCMEC/D3 cells and in NHP BMECs, respectively. Results are given as $PC_R$: quantity T4h basolateral / quantity T4h total (apical + basolateral + intracellular). For pulse chase ratio analysis, a superiority analysis was conducted for each antibody using a one sample one-sided t-test analysis versus constant. The superiority margin was defined as 0.1 for pulse chase ratios. To control multiplicity at 2.5% level, the p-values were adjusted using the Bonferroni-Holm correction. (D) Diagram of a transcytosis experiment. (E) Transcytosis in NHP BMECs. Results are given as the ratio $P_{app}$ test antibody/ $P_{app}$ control antibody. For Papp ratio, a superiority analysis was conducted using an analysis of variance (ANOVA) to compare each antibody versus negative control followed by one-sided Dunnett's test to control multiplicity at 2.5% level. Prior to statistical analysis, data were subjected to inverse transformation to ensure normality of residuals. (B, C, E) Blue bars represent test antibodies, yellow bar represents the reference anti-TFRC antibody. For each parameter, only antibodies with at least 3 measurements were included in statistical analysis. NS: p-value>0,025, *: 0,01< p-value <0,025, **: 0,001< p-value <0,01, ***: p-value <0,001 (A and D) images created by Biorender.

After 4 hours of chase the amount of antibody was measured in each compartment and the "*pulse-chase*" ratio ($PC_R$) was calculated and used to rank the antibodies. The results are presented in Fig 2B. Most of the monoclonal antibodies demonstrated higher transcytosis capacity in the "*pulse-chase*" assay with hCMEC/D3 cells than the reference anti hTFRC antibody, with the best ones being 4F2 and 6D6 Fig 2B). Two antibodies turned out to be negative (4D2 and 3E8). As expected, 8C12 did not undergo transcytosis since it did not bind to hCMEC/D3 cells.

**Primary NHP BMECs *pulse-chase* experiments.** The same set of monoclonal antibodies was evaluated in a "*pulse-chase*" assay in NHP BMECs, using the same protocol as in hCMEC/D3 cells. The results are outlined in Fig 2C.

4F2 came out again as the best candidate. In this assay 8C12 was found to undergo transcytosis in NHP cells while it did not in hCMEC/D3 cells, confirming the lack of human/NHP cross-reactivity for this antibody. 3E8 did not undergo transcytosis in NHP BMECs or in hCMEC/D3 cells. 6F5, 6D6 and 4D2 behaved differently in the two models despite binding to both NHP and human cell lines.

**Transcytosis experiments with primary NHP BMECs.** Finally, the antibodies were evaluated in a model of transcytosis in primary NHP BMECs. The model reported by Chaves et al. [33] (Fig 2D) is well characterized, with high TEER values and tight junctions; it is conducted in Transwell® inserts and the bottom well is cultured with astrocytes. An anti-hTFRC antibody showed high $P_{app}$ compared to a control antibody in this model [53].

A mixture of the monoclonal antibodies and a mouse IgG, used as a control to evaluate permeability of the cell layer, was introduced into each well in the apical compartment and quantified in the basolateral compartment using ELISA. In this model the monoclonal antibodies were ranked based on the ratio of their apparent permeabilities *versus* the permeability of the mouse control antibody $P_{app}$ ratio = ($P_{app}$ test antibody / $P_{app}$ control antibody). The results are shown in Fig 2E.

Once again 4F2 came out as the best candidate, although in this assay all the antibodies appeared to undergo less transcytosis than the anti-TFRC antibody. 3B8, 3C5 and 9F4 performed as in the two previous models, showing good to moderate levels of transcytosis, while 3E8 confirmed poor transcytosis. All Transwell® inserts in which the Papp value of the control mouse IgG was over 150% of the median control $P_{app}$ were eliminated. This was the case for 8C12 for which we were not able to conclude on its transcytosis capacity in this NHP model.

## Target identification

Two complementary methods were used to identify the targets of the antibodies: either deconvolution by immunoprecipitation with a lysate of hCMEC/D3 cells (IP)/LC-MS and confirmation by ELISA with recombinant protein, or cell microarray technology at Retrogenix™/Charles River. The results are reported in Table 2. Full Retrogenix™/Charles River cell microarray results are shown in S2 Fig.

Aside from 1E7 clone binding to ALCAM, 6C7 clone binding to Striatin3, 6A10 and 3E8 for which no antigen was found, all the other antibodies recognized integrins in at least one deconvolution method. The best candidate 4F2 recognized the integrin β1 subunit combined with any α subunit, while several other hits bound to α5β1 heterodimers.

To elucidate the potential of these receptors in mediating transcytosis we then explored which subunit of the integrin was involved, the level of affinity needed for optimal transcytosis and the function of the antibody in the integrin signaling cascade.

**Table 2. Antibody target identification.**

| clone | Target Identification with IP/LC-MS | Human soluble protein confirmed by ELISA following IP/MS | Target Identification and confirmation of human membrane protein with Retrogenix™ cell microarray |
|---|---|---|---|
| 1E7 | ALCAM | ALCAM | |
| 3B8 | No protein found | | ITGA5 + ITGB1 |
| 3C5 | ITGAV, ARPC5, ITGB3, ITGB5, SRSF1 | | ITGAV + ITGB1, 3, 5, 6, 8 |
| 3E8 | LAMP1, SP16H, RCN-1 | | |
| 4D2 | ENPL | | ITGA5 + ITGB1 |
| 4F2 | ITGA5, IGTA3, ITGB1 | ITGA5 + ITGB1 | ITGAn + ITGB1 |
| | | ITGA3 + ITGB1 | |
| 6A10 | No protein found | | |
| 6C7 | Striatin3 | Striatin3 | no target identified, Striatin3 not found |
| 6D6 | RSSA | | ITGA5 + ITGB1 |
| 6F5 | ITGA5, ITGB1 | ITGA5 + ITGB1 | No integrin heterodimer confirmed |
| 8C10 | ITGA5, ITGB1 | ITGA5 + ITGB1 | ITGA5 + ITGB1 |
| 8C12 | ITGA5, ITGB1 | ITGA3 + ITGB1 weak binding | No integrin heterodimer confirmed* |
| 9F4 | No protein found | ITGA5 + ITGB1 | ITGA5 + ITGB1 |

*8C12 did not show binding to human hCMEC/D3 cells

## Binding of identified monoclonal antibodies to several mouse and human integrin subunits

The ability of the antibodies to bind human α3, α4, α5 and β1 monomer subunit, human α4β1 dimer and mouse and human α3β1 and α5β1 dimers was assessed using ELISA, with commercially available recombinant integrin subunits. All the above monoclonal antibodies were assessed, as well as several commercially available anti-α5, α3 or β1 integrin antibodies, along with natalizumab [54] a therapeutic anti-α4β1 antibody marketed in multiple sclerosis and OS2966, an integrin β1 blocking antibody [55, 56].

All commercially available anti β1-integrin antibodies tested (entries 1–9), except MA517103, entry 8, displayed binding to the β1-integrin subunit as expected and to all its heterodimers α3β1, α4β1 and α5β1. MA517103, which is described as an anti β1-integrin only reacted with α3-integrin, albeit with modest affinity. The anti-human α3-integrin antibodies (entries 10–14) bound to either the α3 monomer and/or α3β1 heterodimers with 343802 (entry 11) showing no selectivity over other subunits while the anti α3β1 antibody MAB1346 (entry 13) confirmed binding to this heterodimer. Finally, anti α5-integrin MFR5 and anti α4-integrin natalizumab antibodies (entries 15 and 16, respectively) confirmed binding to the α5β1 and α4β1 heterodimers respectively with no binding to the α monomers (Table 3).

Of our internally produced antibodies, only three displayed binding, 4F2 (entry 17) to α3 and β1 monomers and α3β1, α4β1 and α5β1 dimers confirming the character of pan α β1 targeting (αnβ1) as found during the target identification experiments, while 6F5 and 8C10 (entries 18,19) only bound to α5β1. The remaining internally generated antibodies did not display binding to any monomers or heterodimers in these conditions. This could be linked to lower affinity or recognition of a cell-exposed conformational epitope that does not exist in the recombinant soluble proteins.

Altogether, the profile of our best candidate 4F2 (entry 17) was very similar to the one of OS2966 (entry 4) and others like MABT409, MABT199 and 14-0299-82 (entries 3–5,7) described as β1 subunit binders while 6F5 and 8C10 (entries 18–19) displayed specific binding to the α5β1 heterodimer, like MFR5 (entry 15).

**Table 3. Anti-integrin antibodies apparent affinities to different integrin subunits investigated by ELISA.**

| Entry | Reference | Commercial target | Function[a] | Cross-reactivity[b] | EC$_{50}$ in nM | | | | | | | | |
|---|---|---|---|---|---|---|---|---|---|---|---|---|---|
| | | | | | human α3 | human α4 | human α5 | human β1 | human α3β1 | human α4β1 | human α5β1 | mouse α3β1 | mouse α5β1 |
| 1 | MAB1965 | β1 | Inhibitory | human | Nb | Nb | Nb | 0.04 | 0.02 | 0.03 | 0.02 | Nb | Nb |
| 2 | NBP2-52708 | β1 | Non-functional | human | Nb | Nb | Nb | 0.16 | 0.19 | 0.25 | 0.13 | Nb | Nb |
| 3 | MABT409 | β1 | | Human, Bovine, Pig, Sheep, Horse, Canine, Rhesus Macaque | 0.60 | Nb | Nb | 0.73 | 0.65 | 0.59 | 0.38 | Nb | Nb |
| 4 | OS2966 | β1 | Inhibitory | human | 3.30 | Nb | Nb | 0.60 | 0.40 | 0.70 | 0.80 | Nb | Nb |
| 5 | 14-0299-82 | β1 | Stimulatory | human mouse | 0.38 | Nb | Nb | 1.30 | 0.50 | 0.68 | 0.80 | Nb | Nb |
| 6 | MAB2079Z | β1 | Stimulatory | rat human | Nb | Nb | Nb | 3.55 | 3.00 | 1.06 | 0.88 | Nb | Nb |
| 7 | MABT199 | β1 | Stimulatory | human | 4.18 | Nb | Nb | 2.24 | 1.32 | 0.75 | 0.97 | Nb | Nb |
| 8 | MA5-17103 | β1 | | human, mouse, NHP | 10.88 | Nb | Nb | Nb | Nb | Nb | Nb | Nb | Nb |
| 9 | 553715 | β1 | Stimulatory | mouse | Nb | Nb | Nb | 0.10 | 0.30 | 0.20 | 0.30 | 0.20 | 0.70 |
| 10 | MA1-25298 | α3 | Non-functional | human | 15.90 | Nb | Nb | Nb | Nb | Nb | Nb | Nb | Nb |
| 11 | 343802 | α3 | | human | 0.70 | Nb | Nb | Nb | 0.17 | Nb | Nb | Nb | Nb |
| 12 | 66070-1-Ig | α3 | | human, pig, mouse | 28.10 | 8.94 | 6.40 | Nb | 0.48 | 14.00 | 5.30 | 0.26 | 5.30 |
| 13 | MAB1346 | α3 | Inhibitory | human | | | | | 0.06 | | | Nb | |
| 14 | DCABH-8217 | α3β1 | | human | | | | | 0.29 | | | | |
| 15 | MFR5 | α5 | | mouse | Nb | Nb | Nb | Nb | Nb | Nb | 0.10 | Nb | Nb |
| 16 | Natalizumab | α4 | Inhibitory | human | Nb | Nb | Nb | Nb | Nb | 0.17 | Nb | Nb | Nb |
| | | α4β1 | | | | | | | | | | | |
| | | α4β7 | | | | | | | | | | | |
| 17 | 4F2 | NA | | human, NHP | 3.00 | Nb | Nb | 0.60 | 0.20 | 0.40 | 0.20 | Nb | Nb |
| 18 | 6F5 | NA | | human, NHP | Nb | Nb | Nb | Nb | Nb | Nb | 1.00 | Nb | Nb |
| 19 | 8C10 | NA | | human, NHP | Nb | Nb | Nb | Nb | Nb | Nb | 1.50 | Nb | Nb |
| 20 | 3B8 | NA | | human, NHP | Nb | Nb | Nb | Nb | Nb | Nb | Nb | Nb | Nb |
| 21 | 3C5 | NA | | human, NHP | Nb | Nb | Nb | Nb | Nb | Nb | Nb | Nb | Nb |
| 22 | 4D2 | NA | | human, NHP | Nb | Nb | Nb | Nb | Nb | Nb | Nb | Nb | Nb |
| 23 | 6D6 | NA | | human, NHP | Nb | Nb | Nb | Nb | Nb | Nb | Nb | Nb | Nb |
| 24 | 8C12 | NA | | NHP | Nb | Nb | Nb | Nb | Nb | Nb | Nb | Nb | Nb |
| 25 | 9F4 | NA | | human, NHP | Nb | Nb | Nb | Nb | Nb | Nb | Nb | Nb | Nb |

a: As reported in Byron et al [57]

b: As reported by commercial provider; NA: not applicable; Nb: non binder

In order to validate the mechanism *in vivo*, we would need to identify a mouse anti-integrin antibody performing transcytosis. None of our internally generated antibodies displayed affinity for any of the murine subunits. Interestingly, 553715 (entry 9), which was generated from a mouse endothelial cell line and not reported to be cross-reactive, also showed binding to the human subunits. 66070 (entry 12) confirmed its mouse-human cross-reactivity. These were selected for evaluation in a transcytosis model.

Anti-integrin antibodies have been reported to bind to integrin in different conformations corresponding to several functions [57]: stimulatory or activation-specific, inhibitory, or non-

functional. To assess the potential impact of this functionality conformation on transcytosis, we tested a few commercially available anti-integrins with such described functions in our transcytosis models.

Aiming to clarify which subunit could be at stake, whether the function of the antibody was important and whether the mouse-human cross-reactive commercially available antibodies were able to perform transcytosis, we performed a new set of transcytosis experiments in a NHP *in vitro* model.

### Transcytosis assessment in the PharmaCo-Cell® NHP primary model

This commercially available model uses three cell types. Transwell® inserts are coated with primary cultures of NHP (*Macaca irus*) BMECs, brain pericytes and astrocytes. This BBB Kit™, reconstituted by triple co-culture, shows BBB features such as TEER values around 180 $\Omega.cm^2$. This model is comparable to the one used above [33] and has the advantage of being commercially available and therefore available upon need. Several of the above monoclonal antibodies were assessed in this model (Fig 3A).

The two commercially available anti-β1 integrin antibodies 66070-1-Ig and 553715 which displayed binding to both human and mouse integrin subunits were evaluated in the PharmaCo-Cell® model with hCMEC/D3 cells to validate transcytosis in a human model. The results are shown in S3 Fig. As the antibody 553715 displayed a much higher $PC_R$ in the hCMEC/D3 *pulse-chase* assay than antibody 66070-1-Ig, it was selected for further evaluation in the PharmaCo-Cell® NHP transcytosis model. However, it did not cross the Transwell®, eliminating our hope to use it for *in vivo* validation. There was no difference between anti β1 antibodies with inhibitor, activator or no function and most of them displayed no or very little transcytosis. The only antibody that underwent transcytosis to some extent was OS2966. In this assay 4F2 behaved like the anti-TFRC monoclonal antibody, leading to the highest

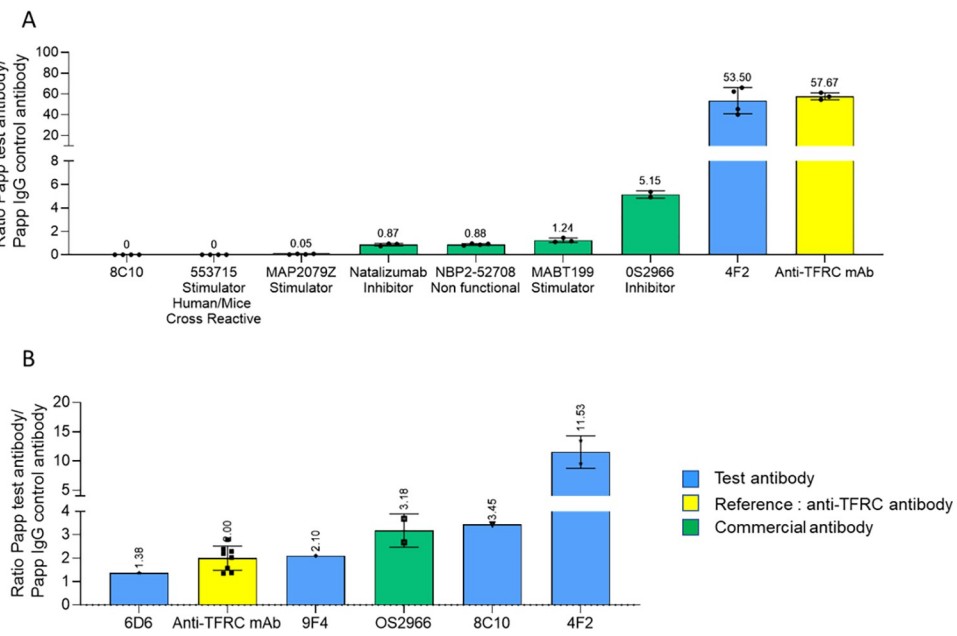

**Fig 3. Antibody transport in two commercial *in vitro* BBB models.** (A) Transcytosis in the PharmaCo-Cell® Transwell® model. (B) Transcytosis in the Brainplotting™ Transwell® model. Results are given as $P_{app}$ test antibody/ $P_{app}$ control antibody. Blue bars represent test antibodies, the green bar represents a commercial antibody, and the yellow bar represents the reference anti-TFRC antibody. Each point represents one Transwell®.

transcytosis. In comparison, commercial integrin antibodies were poorly active in transcytosis, an observation that highlights the originality of the novel antibodies discovered here.

## Evaluation of the best candidates in a human primary model of transcytosis

We evaluated several of our antibodies with different binding profiles in a primary human BMEC Transwell® model from Brainplotting™ [35]. This model is derived from primary human BMECs prepared from brain samples obtained after surgical resection. The results are shown in Fig 3B.

In this model, OS2966, 4F2 and 8C10 underwent transcytosis more efficiently than the anti TFRC monoclonal antibody. Antibody 9F4 was comparable to anti TFRC, while 6D6 was less efficient. These results confirm the superiority of 4F2 to undergo transcytosis.

We verified that the results of our transcytosis assays were not linked to increasing the permeability of our models. If this were the case, the controls should also appear with higher apparent permeability. We compared $P_{app}$ of all the control antibodies incubated in each Transwell® insert with those of Transwell® inserts incubated with the anti-TFRC antibody (Fig 4A–4C). We also checked the integrity of the tight junctions with fluorescein transport evaluation (Fig 4D), TEER measurements (Fig 4E) and zona occludens (ZO)-1 immunofluorescent labeling at the end of the transcytosis experiment with human BMECs from Brainplotting™ (Fig 4F).

All control antibodies display similar $P_{app}$ to $P_{app}$ of control antibody and fluorescein in cells incubated with anti-integrin and anti-TFRC antibodies, suggesting no effect on cell permeability (Fig 4A–4D). In Brainplotting™ model, TEER measurement in Transwell® treated with 4F2 or anti-TFRC were similar (Fig 4E) and ZO-1 labelling suggests that tight junctions were not disrupted by incubation with antibodies (Fig 4F). We have also stained hCMEC/D3 with 4F2 antibody. The S6 Fig shows 4F2 target is present on the cell membrane.

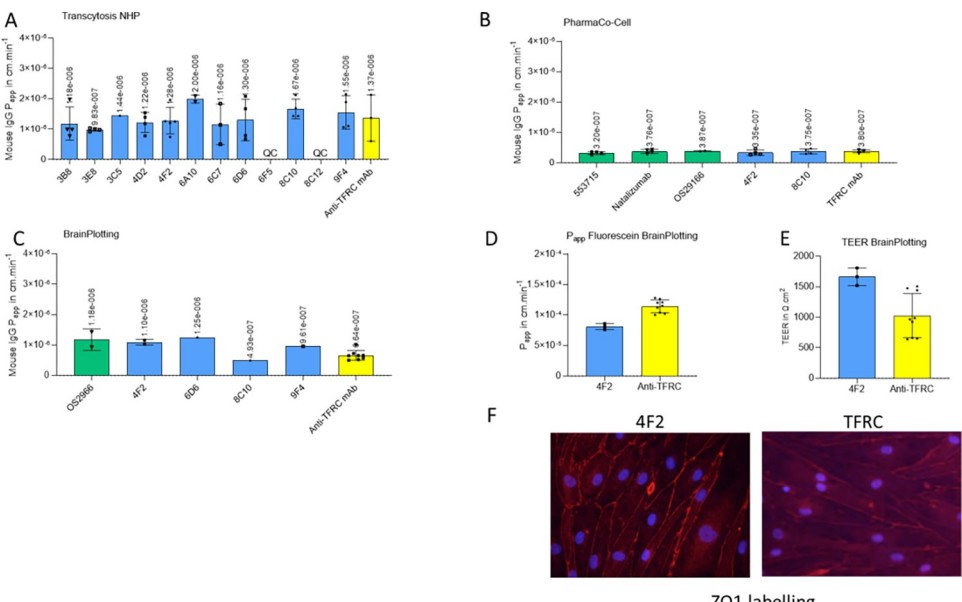

**Fig 4. All mouse IgG control $P_{app}$ in different transcytosis models.** (A) NHP internal model, (B) Human Brainplotting™ model, (C) NHP PharmaCo-Cell® model, (D) Fluorescein $P_{app}$ in 4F2 and anti-TFRC Transwell® (E) TEER measurements with Evohm instrument. (F) Human BMEC immunofluorescence ZO1 labelling in red and nuclei with Hoechst in blue after incubation with antibodies in Brainplotting™ model. Each point represents one Transwell®.

## Immunohistochemistry on human and non-primate tissues

In parallel to these efforts, we undertook immunohistochemical evaluation of the monoclonal antibodies on human and non-human primate tissues to further prioritize them. The objectives were to determine binding on these tissues and cellular localization in brain and periphery. A comparison with human TFRC was also made in these experiments. A first screening was performed on PC3 cells to assess if the monoclonal antibodies could be used on paraffin-embedded or on frozen material, then in hCMEC/D3 cells (Fig 5A).

In supplementary data (S4 Fig), only four monoclonal antibodies gave specific binding on frozen human brain tissue: 4F2, 4D2, 3C5 and 3B8. Of these, only 4F2 showed specific vascular staining. 3B8, 4D2 and 3C5 displayed non-vascular staining in human brain tissue while the 3C5 stained the basal membrane of capsule glomeruli, 50% tubules, nerve and other tissues. As for the anti TFRC monoclonal antibodies, 4F2 displayed binding to BMECs. While TFRC was widely expressed on neurons and glia, 4F2 was clearly restricted to parenchyma, choroidal and meninges capillaries, along with arterial smooth muscle, and basement membrane (Fig 5B).

The same pattern of expression could be seen in human and non-human primate tissue (Fig 5C) along with peripheral tissues such as heart, pancreas, kidney, liver.

To assess the 4F2 antibody target localization in microvessels, we performed immunohisto-fluorescence in frozen healthy NHP brain slices (Fig 5D). Images confirmed vascular staining with our 4F2 antibody but did not permit to conclude on a precise localization of 4F2 at the endothelial cell membrane.

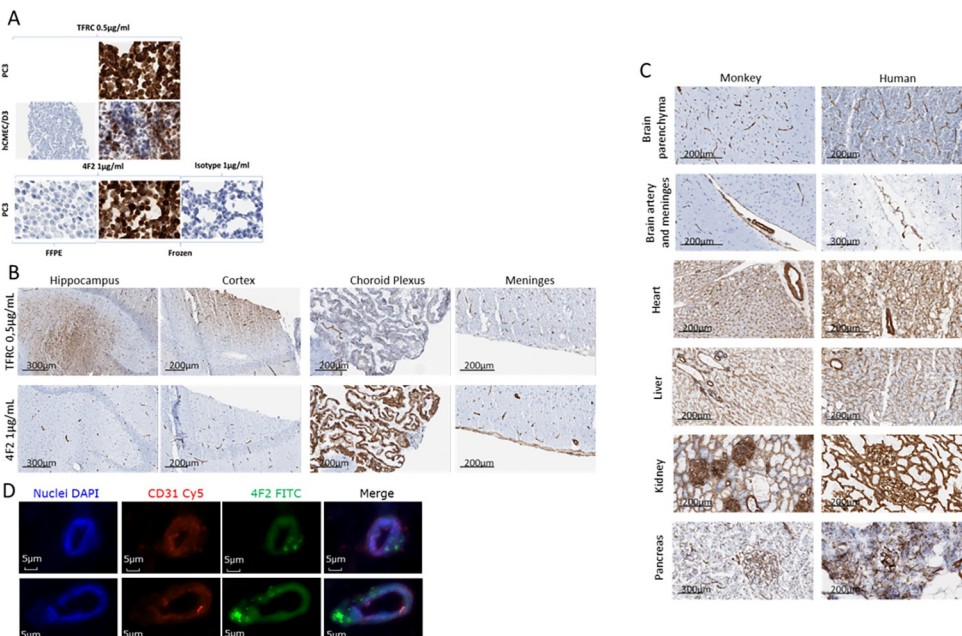

**Fig 5. IHC with anti-TFRC and anti-4F2 antibodies.** The blue staining corresponds to the hematoxylin counterstaining to visualize cells and nuclei. Positive staining are in brown (DAB 3,3'-Diaminobenzidine) (A) IHC in FFPE or frozen cells to select samples. Cells were visualized with a microscope at 200x (B) 4F2 antibody IHC for brain parenchyma specificity staining on NHP brain slices compared to anti-TFRC antibody (C) 4F2 antibody human-NHP cross-reactivity in human and NHP slices of several organs. Isotype controls are in supplementary data (S4 Fig) (D) Fluorescent IHC in NHP brain capillaries from frozen healthy NHP brain slices stained with Dapi (nucleus), Cy5 (anti-CD31 antibody) and FITC (4F2 antibody).

## Discussion

Our initial goal was to generate antibodies that would recognize human receptors, by immunizing mice with primary human BMECs. Due to the very high number of cells required for immunization campaigns, this would have required generating primary cells that could be further multiplied. We show that even when using fresh brain tissue after short post-mortem intervals, the endothelial cell phenotype was largely lost upon culture. In contrast, we were able to develop a robust procedure to prepare primary NHP BMECs of good quality [33]. Given the homology of many proteins between human and NHP, we decided to start from these cells and to include binding to both human and NHP BMECs in the flowchart to select only cross-reactive antibodies. From immunization of Trianni mice with primary healthy NHP BMECs we collected 634 antibodies of which 62 bound to human hCMEC/D3 cells or NHP BMECs. Among the 22 antibodies with unique sequences, 12 clones internalized in NHP BMEC and hCMEC/D3 were further evaluated in three transcytosis models. The most relevant model to evaluate the antibodies from the campaign would be with the same cells as the ones used for immunization, namely the NHP BMECs. This model described by Chaves et al. [33] has been validated with transferrin along with an anti-human/NHP TFRC antibody. The hCMEC/D3 immortalized human brain endothelial cell line [58, 59] has been well characterized in terms of brain-specific and tight junction proteins [60]. Given the TEER values indicative of high paracellular permeability, we used this cell line in a *pulse-chase* mode which allows to avoid the effect linked to non-specific paracellular passage [52]. This procedure was also applied for primary NHP BMECs with TEER $< 150\ \Omega.cm^2$. The best antibody detected in the three models was 4F2.

In parallel, uncovering the targets of the antibodies validated in these three models of transcytosis was performed using two different methods. The first was by immunoprecipitation/mass spectrometry followed by confirmation by ELISA and the second by Retrogenix™/Charles River reverse transfection microarray technology. Of the twenty-two antibodies, one displayed ALCAM as its target, one interacted with Striatin3 and nine bound to β1 integrin subunit or heterodimers.

ALCAM (also known as CD166-antigen) has been reported to play a major role in infiltration of brain tumors by T-cells [61] and is very relevant to the BBB field. We identified one antibody against ALCAM (1E7) in this campaign, but it did not perform well in the transcytosis assay. In addition, antibodies against ALCAM have been reported to inhibit diapedesis of monocytes across the BBB [62]. We decided not to pursue this protein as a target for enhanced brain delivery.

Striatin3 (also known as SG2NA) is a 780 amino-acid protein with four protein-protein interaction domains including a caveolin-binding domain, a coiled-coil domain, a $Ca^{2+}$-calmodulin binding domain and a tryptophan-aspartate (WD)-repeat domain [63–66]. Tissue expression is mostly ubiquitous but regulated in specific tissues by differential splicing [67]. Even if Striatin3 is localized in multiple cellular compartments, plasma membrane, mitochondria, endoplasmic reticulum, nucleus, and lysosomes [68] this protein is essentially intracellular and has no extracellular domain. In addition, the Retrogenix™/Charles River cell microarray technology did not validate this target identified by immunoprecipitation and mass spectrometry. Despite encouraging transcytosis results, we decided not to pursue Striatin3 and to focus on the integrin antibodies.

Integrins form a family of 24 heterodimeric transmembrane receptors that play an active role in cell-cell interactions in a multitude of physiological and disease situations [57]. They are important players in oncology [69], cancer, melanoma and metastatic progression, but also in tissue fibrosis [70], rheumatoid arthritis [71], wound healing [72] and ischemic stroke [73]

to name a few [74]. Integrins play an important role in the adhesion of cells to the extracellular matrix [44, 72, 75]. They have been shown to mediate the endocytosis of a variety of material such as nanoparticles [44, 76–80], beads [81], phages [82, 83] or cells [84, 85] provided that the material presents integrin ligands on its surface. Most known examples have used RGD peptides, canonical ligands of several integrins [86] or their derivatives, but one example reports on anti α2β1 integrin antibody-targeted nanoparticles leading to enhanced cell uptake [79]. Some pathogenic bacteria such as *Y. pseudotuberculosis* [87–90], *E. coli* [91, 92], *S. aureus* [84, 85], staphylococci and streptococci [93, 94], *Neisseria meningitidis* [95], virus [96] or adeno-associated-viruses [97] have also been shown to be internalized in cells via an integrin-mediated pathway. In most of these studies, integrin β1 is the main reported player. Several mechanisms have been documented for this uptake, including macropinocytosis and transcytosis [79, 87, 96], pathways involving clathrin [75] caveolae, dynamin and dynamic unstable microtubules [72, 92]. Uptake has been shown to be dependent upon the integrin isoforms [44] and state of activation [75].

Transcytosis of integrin-ligand-exposing-cells [98, 99], -nanoparticles [100–102], or -bacteria [103] and -viruses [104] has also been reported but most of the literature is on epithelial cells. Very few studies describe transcytosis through monolayers of endothelial cells. Two examples related to brain metastasis have shown 1.3-to-1.5-fold enhanced transcytosis of breast cancer cell lines expressing β1 integrin ligands through monolayers of human brain microvascular endothelial cells [83] and mouse brain endothelial cells [82] respectively. One article reports that Streptococcus group B are using α5β1 and αvβ3 for brain invasion in juvenile meningitis [105]. RGD-decorated nanoparticles have shown modest improvement of transcytosis in the mouse bEnd3 cell model [106] while in another report an enhanced transcytosis could only be observed in a blood tumor model while a BBB model with rat primary brain endothelial cells failed to show an enhancement [107]. All the examples of uptake and transcytosis discussed above are with particles or cells decorated with many integrin-targeting ligands, and transcytosis was improved with increasing ligand density [100]. Although a few of the above examples have shown that uptake or transcytosis can be inhibited in the presence of anti-β1 integrin antibody [100, 105], to the best of our knowledge no single ligand and more specifically no anti-integrin antibody has been reported to be internalized or to undergo transcytosis through any cell or model.

A few anti-integrin monoclonal antibodies are on the market as drugs, such as natalizumab, an anti-α4β1 integrin for multiple sclerosis [108], abciximab and efalizumab [109], an anti-αIIbβ3a integrin for thrombosis and an anti-αLβ2 for psoriasis while others are in clinical development. Relevant to our studies are three anti-β1 antibodies OS2966 (anti-β1) [55], MINT1526A (anti-α5β1) [110] and M100 (volociximab) (anti-α5β1) [110] currently at various stages of clinical development in oncology. None of them has been reported to display brain exposure.

Integrin β1 is described to play a key role in repair and protection of the neurovascular unit during cerebral ischemia [111] and integrin α5 receptor has been linked to barrier tightness [112]. A few studies have analyzed the BBB following application of anti-integrin β1 antibodies. Osada et al. observed that functional inhibition with an anti-integrin β1 antibody resulted in lowered CLDN5 expression and deduced a critical role for β1-integrin-mediated adhesion of brain endothelial cells to the surrounding extracellular matrix for stabilizing CLDN5 in BBB tight junctions and BBB integrity [113, 114]. Their results were confirmed in another study in which CLDN5, occludin and ZO1 were decreased by anti-integrin β1 antibody treatment [115]. In contrast, Li et al didn't observe CLDN5 or ZO-1 reorganization after treatment with anti- β1 antibody [111]. Furthermore, Edwards and Roberts have shown that mice with endothelial cell-specific knockout of α5 integrin display enhanced barrier tightness after stroke or

oxygen-glucose deprivation [116, 117]. It was therefore important for us to make sure that the transcytosis we observed for our anti integrin antibodies was not a consequence of endothelial cell permeabilization. This appeared not to be the case because the control antibodies did not show enhanced permeability in the presence of the anti-integrin antibodies (Fig 4A–4E). Secondly, the tight junctions were still in place at the end of the experiment as shown by ZO-1 immunofluorescence (Fig 4F).

After this validation, we tried to understand which subunit was recognized by our best antibodies and if we could correlate a binding or a specific αβ heterodimer to a transcytotic capacity. For this, binding was performed by ELISA using recombinant α and β integrin subunits (Table 3). This analysis was performed also for a series of commercially available or reported anti-integrin antibodies. Only three of our antibodies displayed binding to the recombinant proteins, with 4F2 binding to α3, β1 and all α3,4,5β1 heterodimers, while 6F5 and 8C10 only bound to α5β1. The others may recognize a specific conformation-dependent epitope of the integrins not present in these recombinant forms. Several commercial antibodies displayed the same profile as 4F2, including OS2966 which is known to bind to β1 [56, 118]. The best antibody for transcytosis, 4F2, bound to more heterodimers but this did not appear to be a sufficient condition, since antibodies with a similar profile such as MABT199 (Table 3, entry 7) did not perform transcytosis (Fig 3A). This did not seem linked to the lower affinity of MABT199 for β1 and its heterodimers since NBP2-52708, which has a higher affinity (Table 3, entry 2) also showed very low transcytosis (Fig 3A).

We then tried to investigate whether the transcytosis could be linked to functionality of the antibody. It is now understood that anti-integrin antibodies can be classified into three main types: those that inhibit ligand engagement, those that stimulate ligand engagement or induce a high-affinity conformation and those that have no specific effect (and behave as negative controls) [57]. These functions are intimately associated with the three potential conformations of integrins (active, at rest and inactive [119]) bound by the antibody. Blystone et al. established for instance that α5β1-mediated phagocytosis required activation to a high affinity conformation by an activating monoclonal antibody. The phagocytosis was inhibited by an excess of its ligand fibronectin [81]. The cognate ligands of α3β1 and α5β1, the most frequent hits of our antibodies are laminin [120] and fibronectin [121] respectively. Transcytosis was not inhibited by the presence of fibronectin, either by Transwell® coating or when added to the medium (S5 Fig) suggesting that our antibodies might not act by inducing a high affinity conformation. To gain insight into the impact of functionality on transcytosis, we selected commercially available anti-integrin antibodies with defined functions, such as MABT199, 553715 and MAP2079Z, all described as activating antibodies and NBP2-52708 described as a non-functional antibody. In contrast, OS2966, like most of the anti-integrin antibodies currently in development [57] has been reported as an inhibitor of downstream signaling from integrin β1 [56]. Evaluation of these antibodies in a NHP primary model (Fig 3A) showed that only OS2966 underwent transcytosis to some extent, albeit much less than 4F2, suggesting that inhibition might be necessary. This transcytosis property is not reported for OS2966, which has in fact been administered intra-tumorally in a mouse model of glioblastoma [55].

4F2, with its best transcytosis efficacy was also the only one to demonstrate specific staining of vessels on human and NHP brain and peripheral tissues suggesting high specificity for endothelial cells. Using electronic microscopy and immunogold labeling, Conforti et al. have shown that integrin receptors such as αvβ3, α3β1, α5β1 and others are not only located to the basal side of endothelial cells, but also on the cell surface in contact with blood [122]. Data indicate, that in addition to their role in promoting extracellular attachment to extracellular matrix proteins, integrin receptors can be exposed to the bloodstream and eventually be available for binding of plasma proteins, circulating cells and antibodies. Our confocal microscope

immunofluorescence study (Fig 5D) did not allow to determine whether this labeling with 4F2 was luminal or abluminal. Co-labeling with Permeability-GlycoProtein (P-gp) in the same tissues could have been informative in this respect [123].

The different rank orders obtained for the antibodies in the five BBB models displayed in this paper illustrate the challenges of such models. The hCMEC/D3 model, as some other cell line-based models, has been shown to lose some of its barrier properties upon culture [18]. Given their low transendothelial resistance, these cells are probably more suited for mechanistic studies than true permeability assessments [124]. For this reason, we have used this model in a "*pulse-chase*" mode [52]. The NHP models used differ by the presence of astrocytes and pericytes which can have a major impact on permeability and protein expression. Pericyte coverage is essential to the barrier integrity [125], moreover pericyte signaling to endothelial cells via integrins has recently been reported to impact BBB permeability [112]. The PharmaCo-Cell® model is certainly the most relevant to this study, as immunization was performed with NHP cells, and this model incorporates cells from the neurovascular unit. Finally, the human primary model is derived from epilepsy or glioblastoma surgical resections and even though they are taken from the healthy surrounding tissue, some variability could be present. The extent to which *in vitro* BBB models can predict what occurs *in vivo* is still the matter of intense research and debate. Even if the brain exposures of some antibodies have been linked to their apparent permeabilities in *in vitro* transcytosis models [126, 127], *in vivo* brain exposure, distribution and pharmacokinetics are dependent on a series of dynamic processes, also involving target engagement, localization and cellular trafficking. To conclude on the potential of integrin receptors to transport antibodies to the brain, we tried to identify a mouse/human reactive antibody for which transcytosis *in vitro* could be validated for direct evaluation *in vivo* in mice. None of our internally generated antibodies turned out to cross-react with mouse integrins, but one of the commercially available antibodies, 553715 [128], demonstrated binding to both human and mice subunits and heterodimers (Table 3, entry 9) suggesting cross-reactivity. However, in the non-human primate *in vitro* transcytosis model 5537715 revealed no barrier crossing in comparison with 4F2 and a reference anti-TFRC antibody (Fig 3A).

We tried to understand the reason for the superiority of 4F2 *versus* all other anti-integrin antibodies. The first potential reason is their pan alpha binding profile but several commercial antibodies displaying the same binding profile did not perform transcytosis. Positively charged proteins are known to display higher tissue and brain penetration [129]; however predicted pIs (S3 Table) did not point to a higher pI for 4F2 compared to the other antibodies.

This first study validated the flowchart and the models while teaching us several lessons to consider for future campaigns. One limitation of our study is that it did not detect any of the receptors known for mediating transcytosis, such as TFRC, INSR, IGF1R, CD98, LRP1R, LDLR to name a few. We don't know the reason for this but speculate that integrins, which are mediating intercellular interactions and binding to the extracellular matrix [130] might have been shielding these receptors. On the other hand, ITGB1 protein abundance cannot be an explanation for the high hit rate of ITGB1-binding antibodies as it is only 2-fold higher than TFRC but lower than some other receptors such as LRP1 or membrane transport ATPases [131]. Integrins are ubiquitously expressed, and so are all the above reported transcytosis-mediating receptors. Integrin α7 antibodies have for instance been reported to increase muscle targeting of lysosomal enzymes [132]. Nevertheless, in our next study, we could include a counter screen of peripheral tissues, to select hits with the highest endothelial brain cell enrichment.

Inherently to the host, it is notoriously difficult to obtain cross-reactive antibodies for highly conserved mouse-human targets during mouse immunization campaigns. Alternative solutions would be to screen naïve antibody libraries against human and mouse brain primary

endothelial cells to select cross-reactive clones or to immunize lama and select cross-reactive nanobodies.

This study is the first one reporting on the potential of BBB crossing for very specific anti-integrin antibodies. Regarding the potential of this mechanism for enhancing the brain exposure of biotherapeutics, several important questions remain. siRNA experiments to suppress α3, α5 and β1 genes could bring light on which integrin subunit is required for transcytosis. Co-structure of 4F2 Fab with β1 integrin could provide clues on the activating or inhibiting conformation for transcytosis [133] and on the specific epitope. From this structure, the potential to perform affinity maturation to get mouse-human cross-reactive antibodies could be assessed. Alternatively, epitope mapping could yield information on the antigen that is recognized and whether it is or not conserved across species. The ultimate goal would be *in vivo* validation of brain enhancement *versus* a control antibody first as such and then as a bispecific antibody or fused to a cargo therapeutic protein. Integrin α5β1 has been reported as a new target for tumor treatment [134]. Given the overexpression of integrins in several tumor cell models [56, 135] and their major role in tumor development [136–138] this mechanism could be extremely useful for the treatment of central tumors such as glioblastoma.

## Supporting information

**S1 Fig. Human primary BMEC production.** [A] Human primary brain cells isolated from NBB autopsy cases at passage 0 after 7 days in culture. Cells were observed 100X magnification with inverted microscope [B] Gene Relative Quantity [RQ] in human primary BMECs characterized by qPCR. Data were normalized to data from cortex homogenate.
(TIF)

**S2 Fig. Screening of Retrogenix™/Charles River integrin heterodimers.** Antibodies were loaded at 5μg/mL and bound to different integrin heterodimers spotted on a slide. The signal was detected by HRP coloration.
(TIF)

**S3 Fig. Result of mouse/human cross-reactive commercial anti-β1 integrin antibodies 66070-1-Ig and 553715 evaluated in *pulse-chase* assay with hCMEC/D3 cells.** Internal anti-integrin antibodies are represented with blue bars, commercial anti-integrin antibodies are represented with green bars, anti-TFRC antibody is represented with yellow bar. Each point represents one Transwell®.
(TIF)

**S4 Fig. IHC in frozen brain and peripheral tissue slices.** Immunostaining was done with DABMap™ chromogenic detection kit where brown color is a binding signal. A: human and NHP frozen cortex slides incubated with three different anti-integrin antibodies B: human frozen brain and peripheral tissue slices incubated with 3C5 or 4F2. C: Isotype control of monkey and human slices from *Fig 5*. Blue color corresponds to the hematoxylin counterstaining to visualize the cells and their nuclei.
(TIF)

**S5 Fig. Fibronectin effect on 4F2 and anti-TFRC PC$_R$ in "*pulse-chase*" assay in hCMEC/D3 cells.** 4F2 is represented with blue bars and anti-TFRC is represented with yellow bars. A: different concentrations of fibronectin were coated on the Transwell®. B: different concentrations of fibronectin were added to the culture medium. Each point represents one Transwell®.
(TIF)

**S6 Fig. 4F2 staining in hCMEC/D3.** 4F2 target cellular localization was visualized by fluorescence microscopy at 200X in hCMEC/D3. Cultured cells were fixed in 4% paraformaldehyde for 15 min, at RT, and blocked in Odyssey LiCor Blocking Buffer.4F2 antibody was incubated overnight at 4˚C and anti-human secondary antibody conjugated with Alexa 594 nm and Hoechst 33432 for nuclei staining was subsequently used for 2h at RT. 4F2 target is visualized in red and nuclei in blue.
(TIF)

**S1 Table. Statistical results for pulse chase ratio.**
(DOCX)

**S2 Table. Statistical results for transcytosis ratio.**
(DOCX)

**S3 Table. Total protein, heavy chain and light chain pI calculated for test antibodies.**
(DOCX)

## Acknowledgments

We thank Marc Gumbleton and Jack Smith from Cardiff University for their helpful training on the "*pulse-chase*" assay. We are grateful to Adrienne Peretti-Renaud, Nicolas Maestrali, Stéphane Somarriba, Nathalie Couteault, Laurent Maton, Melanie Annat, David Bournizel, Marie Reau and Evelyne Deschamps from Sanofi for the immunization, hybridoma screening and antibodies production. We are grateful to the Ebiology team from Sanofi for sequence annotations, Stephan Mathieu from Sanofi for his expertise and helpful technical assistance in conducting fluorescent IHC studies, Paola Fiorentini from Sanofi for her investigations to find human brain samples and Isabel Ann Lefevre from Sanofi for English language editing and proofreading the manuscript. We thank Matthias Wabl from TRIANNI Inc and Jim Freeth, Diogo Rodrigues and Jo Soden from Retrogenix™/Charles River.

## Author Contributions

**Conceptualization:** Céline Cegarra, Béatrice Cameron, Catarina Chaves, Tuan-Minh Do, Valérie Roudières, Yi Shi, Dominique Lesuisse.

**Data curation:** Céline Cegarra, Tarik Dabdoubi, Tuan-Minh Do, Bruno Genêt, Valérie Roudières, Patricia Tchepikoff, Dominique Lesuisse.

**Formal analysis:** Céline Cegarra, Béatrice Cameron, Catarina Chaves, Tuan-Minh Do, Bruno Genêt, Valérie Roudières, Yi Shi, Patricia Tchepikoff, Dominique Lesuisse.

**Investigation:** Céline Cegarra, Béatrice Cameron, Catarina Chaves, Tarik Dabdoubi, Tuan-Minh Do, Bruno Genêt, Valérie Roudières, Yi Shi, Dominique Lesuisse.

**Methodology:** Céline Cegarra, Béatrice Cameron, Catarina Chaves, Tarik Dabdoubi, Tuan-Minh Do, Bruno Genêt, Valérie Roudières, Yi Shi, Dominique Lesuisse.

**Project administration:** Céline Cegarra, Béatrice Cameron, Tuan-Minh Do, Dominique Lesuisse.

**Resources:** Céline Cegarra, Catarina Chaves, Tuan-Minh Do, Yi Shi, Patricia Tchepikoff, Dominique Lesuisse.

**Supervision:** Céline Cegarra, Tarik Dabdoubi, Bruno Genêt, Dominique Lesuisse.

**Validation:** Céline Cegarra, Béatrice Cameron, Catarina Chaves, Tarik Dabdoubi, Tuan-Minh Do, Bruno Genêt, Valérie Roudières, Yi Shi, Patricia Tchepikoff, Dominique Lesuisse.

**Visualization:** Catarina Chaves, Tarik Dabdoubi, Tuan-Minh Do, Bruno Genêt, Valérie Roudières, Yi Shi, Dominique Lesuisse.

**Writing – original draft:** Céline Cegarra, Béatrice Cameron, Dominique Lesuisse.

**Writing – review & editing:** Céline Cegarra, Béatrice Cameron, Catarina Chaves, Tarik Dabdoubi, Tuan-Minh Do, Bruno Genêt, Valérie Roudières, Yi Shi, Patricia Tchepikoff, Dominique Lesuisse.

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
