## [Decision Letter · Decision Letter 0]

14 Jun 2022

PONE-D-22-11645An innovative strategy to identify new targets for delivering antibodies to the brain has led to the exploration of the integrin familyPLOS ONE

Dear Dr. Cegarra,

Thank you for submitting your manuscript to PLOS ONE. After careful consideration, we feel that it has merit but does not fully meet PLOS ONE’s publication criteria as it currently stands. Therefore, we invite you to submit a revised version of the manuscript that addresses the points raised during the review process.

Two experts evaluated the work and found it interesting and potentially important, however, many questions were raised which should be addressed, especially those related the BBB specificity and the transcytotic mechanism. 

We look forward to receiving your revised manuscript.

Kind regards,

Mária A. Deli, M.D., Ph.D.

Academic Editor

PLOS ONE

Journal Requirements:

2. In order to comply with PLOS ONE's guidelines for non-human primate experiments (http://journals.plos.org/plosone/s/submission-guidelines#loc-non-human-primates), please provide additional details regarding housing conditions, feeding regimens, environmental enrichment, and all relevant steps taken to alleviate suffering (anesthesia, analgesia, details about humane endpoints, euthanasia, etc.). Also indicate how often animal care staff monitored the health and well-being of the animals and the criteria used to make such assessments. Lastly, specify the disposition of animals at the end of the study (e.g. euthanasia, returned to home colony, etc.). If animals were euthanized following the study, please provide the method of sacrifice.

"All authors are Sanofi employees and may hold shares and/or stock options in the company"

Reviewers' comments:

Reviewer's Responses to Questions

**Comments to the Author**

1. Is the manuscript technically sound, and do the data support the conclusions?

Reviewer #1: Yes

Reviewer #2: No

2. Has the statistical analysis been performed appropriately and rigorously? 

Reviewer #1: N/A

Reviewer #2: No

3. Have the authors made all data underlying the findings in their manuscript fully available?

Reviewer #1: Yes

Reviewer #2: No

4. Is the manuscript presented in an intelligible fashion and written in standard English?

Reviewer #1: Yes

Reviewer #2: Yes

5. Review Comments to the Author

Reviewer #1: In this interesting paper from Cegarra et al., the authors try to find new targets form brain penetrations with a novel strategy. Instead of identifying transcytotic receptors in brain endothelial cells, an immunization with proteins from the endothelial cells is performed. Subsequent transcytotic antibody is selected due to their internalization and transcytose capacity. The paper is base on large work with many experiments, which should be acknowledge. The work is done properly, but perhaps some of the interpretations might be too conclusive. As such I don't have any major issues with the paper, but some suggestions that perhaps would improve the paper.

First of all, EC50 values is in my opinion not data that can measure accurate affinity, more a measurement that allows calculate the antibody concentration needed for a cell/tissue interaction. Kd can not be extracted. (Line 11 ff; Thus, a second objective of the present work has been to investigate the link between the various parameters such as affinity, α and β subunit specificity, functional activity of the monoclonal). Furthermore, the term “Antibody Binding Capacity” is not well introduced. Some elaboration could be beneficial.

Second, the binding of 42F is likely to be subunits of integrins as demonstrated with binding to recombinant protein and pull down. But it does not exclude that it binds to other receptors/proteins. A simple WB could be interesting to demonstrate how specific the 4F2 is.

Regarding internalization; Line 283-84 “ the fluorescence of cells was analyzed with the ImageStream multispectral imaging flow cytometer using the Internalization feature” . The internalization score is an important factor for fig 1C/ selection, but is not clear how it is done. Material and methods are just referring to Internalization feature, but what kind of algorithmic and criteria is behind this? Uptake in endothelia cells grown on slide or Transwells could be highly supportive. E.g in Fig 4D. Why the authors not co-stain against TfR and 4F2 with the Zo staining? Could give important information of cellular localization and thereby sorting mechanism.

(Why is so many control antibodies transcytosed in the carton (Fig 2D)?)

Tightness; ZO1 staining is nice, but is not a valid method to demonstrated tightness. Control antibody is included in most transcytosis experiments, but the result is give as ratios. The Papp would have been valuable for this judgement. And the TEER values is not supportive for the cells being tight.

Normally there is a background in qPCR. Is there no background on e.g. GFAP in qPCR fig 1b? is the primers verified?

Line 438 “The results in Fig 1B show that even if the cells exhibit the right gene profile upon first isolation (P1D0) with high expression of CD31, CLDN 5 and BCRP and no..” what is the right gene profile? It is a few select genes know to be expressed in BEC, but many more (as well as polarity and efflux capacity) should be characterized to demonstrated it is a proper BEC.

It is an interesting approach to immunizes with BEC lysate; it does not find targets that is preferential expressed in brain endothelial cells. The data in fig 5 underscores that there might be a high distribution to other tissues. Do the authors think that this target would be relevant for clinical trials - discuss?

It was questioned if 4F2 binds to luminal and or abluminal membrane (p 822-24 Our confocal microscope immunofluorescence study (Fig 5D) did not allow to determine whether this labeling with 4F2 was luminal or abluminal). An easy experiment would be to use a Transwell NHP cells in a tight monolayer and the add the antibody respectively to each side and follow endocytosis with IF (e.g. 1 hour chase and fix and IF). This would answer the question which is highly relevant.

Reviewer #2: Summary

Overall, this manuscript describes a robust method of producing and identifying BBB targeting antibodies using a mouse immunized with NHP BMECs. This screen identified a highly enriched population of integrin binding antibodies which are suggested to undergo transcytosis. The authors use multiple different in vitro models of human, NHP, and immortalized human BBB models to verify the antibodies transcytosis ratio compared to an internal control. Comparing their lead candidates to integrin antibodies currently on the market or in development to identify possible methods of action. While they were unable to show the method of transcytosis, or if it is able to carry and deliver cargo across the endothelium, this work identified integrins as a potential target for BBB delivery for the first time. The work, while interesting, is lacking statistical support for the conclusions and confirmation that integrin binding is leading to transcytotic or at least vesicle based trafficking.

Major comments

Figure 5 is difficult to interpret as a whole. The figure legend is not descriptive enough to interpret the data being presented. For instance, adding information of the color indications to the caption would be beneficial. Similarly, the complete lack of any scale bars or indications of the size of the field of view make it difficult to discern what the structures are that we are observing, 5D has scale bars, but the size of the scale bars is never indicated. The isotype column in 5C was never described, it’s unclear what tissue this is in and why the most important brain isotype controls are lacking in this figure. These should be included.

Statistical testing to support conclusions is completely absent. There needs to be description of statistical tests in the methods and confidence (p-values) given in the figure legends and/or figures. The form of replication and number of replicates also needs to be reported.

What is the method of transcytosis of these integrin antibodies given that integrins do not typically function in a vesicle trafficking modality? In the discussion the authors discuss the use of integrin β1 blocking antibodies which would be helpful data to include to justify the conclusions. High resolution imaging would also be useful to show co-localization of the integrin and targeting antibody within cellular endosomal/lysosomal/etc structures.

One of the downfalls of this method seems to be the lack of murine cross-reactive antibodies. As these were raised in an immunized mouse model this is not terribly surprising. The authors should discuss how the platform presented can 1. Lead to antibodies that are difficult to test preclinically and 2. Not lead to antibodies against conserved targets between mouse and NHP.

As the authors point out these antibodies seem to be targeting pan-endothelial markers, and one of the problems mentioned in the introduction is the lack of brain specific targeting therapeutics. Could the authors comment on if there is anything known about brain specific or brain enriched integrin heterodimers that could be investigated for this purpose? If as expected, there is not evidence of brain integrin isoform enrichment, the authors should at least speak to the impact of targeting a pan-endothelial protein for CNS drug delivery.

Minor comments

The human primary BMEC data included in figure 1A and B should be removed as these cells are not used in any subsequent panel in the paper. Instead, it would be more beneficial to include similar validation data for the NHP BMECs that were actually used for immunization and include these data in the supplement or new Figs 1A and 1B, even if this data is essentially all represented in the Chaves et al paper. E.g. characterization of the cells actually used to immunize in this paper.

While the authors affirm that the NHP model has good barrier properties, it would be extremely beneficial to describe some of the important characteristics of this model. Particularly as in line 491 the NHP cells were described as not having strong tight junctions and justified the use PCR, but in line 519 these are apparently tight enough to do a traditional transcytosis assay.

Figure 1A needs to have a scale bar indicating the size of the cells.

Figure 1C contains a red squiggle indicative of a misspelled word on the y-axis. This graph would also benefit from pointing out the data points that correspond to the images selected.

How were the internalization score bins determined? And why are absolute internalization scores not reported in table 1 rather than the low/high bins?

Line 644 has an unpaired parenthesis around Figure 4A-C

How does TJ morphology and ZO-1 localization and expression compare to before incubation with antibodies? (Fig 4D). The authors should include the raw Papp data for the control antibodies as described in line 633-634. This is very important to exclude integrin binding as disruptive.

Lines 87-90 speak to the novelty and importance of immunized libraries. The authors should cite a very recent paper from Lajoie et al. scientific reports, 2022 (https://doi.org/10.1038/s41598-022-09962-8) where an analogous strategy was deployed for CNS targeting purposes.

6. PLOS authors have the option to publish the peer review history of their article (what does this mean?). If published, this will include your full peer review and any attached files.

Reviewer #1: **Yes: **Morten S. Nielsen

Reviewer #2: No

---

## [Author Response · Author response to Decision Letter 0]

19 Jul 2022

Reviewer #1: In this interesting paper from Cegarra et al., the authors try to find new targets form brain penetrations with a novel strategy. Instead of identifying transcytotic receptors in brain endothelial cells, an immunization with proteins from the endothelial cells is performed. Subsequent transcytotic antibody is selected due to their internalization and transcytose capacity. The paper is base on large work with many experiments, which should be acknowledge. The work is done properly, but perhaps some of the interpretations might be too conclusive. As such I don't have any major issues with the paper, but some suggestions that perhaps would improve the paper.

First of all, EC50 values is in my opinion not data that can measure accurate affinity, more a measurement that allows calculate the antibody concentration needed for a cell/tissue interaction. Kd can not be extracted. 

We agree with the reviewer. EC50 cannot be called affinity constant (KD) as they are highly dependent on the measurement conditions. Only when the relationship between receptor occupancy and response is linear, can the hypothesis be made that EC50 = KD. EC50 are useful for comparing the relative binding affinities of series of compounds which was done here. To address the reviewer comment, we have replaced affinity by binding or apparent affinity in several instances of the document. In addition, we have inserted the definition of EC50 in the text (lines 300-302): “EC50 is the half maximal effective concentration representing the apparent affinity of the antibody to its target”.

Thus, a second objective of the present work has been to investigate the link between the various parameters such as affinity, α and β subunit specificity, functional activity of the monoclonal). Furthermore, the term “Antibody Binding Capacity” is not well introduced. 

To address this comment, we have indicated the abbreviation ABC when we first describe antibody binding capacity in the material and method section (line 307) and replaced the sentence lines 547-550 “The Antibody Binding Capacity (ABC), which reflects the number of binding sites at the cell surface occupied by the antibody” by “ABC is the number of monoclonal antibodies a sample will bind, and correlates to the number of antigens expressed on the cell surface. It was determined by measuring the binding by flow cytometry. The value was correlated to the receptor binding density using the calibration curve as described in Materials and Methods.”

Second, the binding of 42F is likely to be subunits of integrins as demonstrated with binding to recombinant protein and pull down. But it does not exclude that it binds to other receptors/proteins. A simple WB could be interesting to demonstrate how specific the 4F2 is. 

A WB is a good suggestion. Unfortunately, this antibody doesn’t work in denaturating condition such as WB. To the question of how specific is the 4F2 clone to integrins subunits, we can rely on peptide mapping results after immunoprecipitation, that exhibit a vast majority of peptides (in terms of number of peptides and intensities) belonging to several subunits of integrins, and only 1% of the signal could be addressed to a minor protein, CD151, that has been detected (see table below).

Peptides Intensities count per second (cps) 

Negative ctrl 4F2 Clone Number of Unique 

peptides Protein names

0.00E+00 2.02E+07 2 CD151 antigen

0.00E+00 1.13E+08 7 Integrin alpha-2

0.00E+00 6.19E+08 12 Integrin alpha-3

0.00E+00 3.71E+08 14 Integrin alpha-5;Integrin alpha-5 heavy chain;Integrin alpha-5 light chain

0.00E+00 1.75E+09 14 Integrin beta-1

Regarding internalization; Line 283-84 “ the fluorescence of cells was analyzed with the ImageStream multispectral imaging flow cytometer using the Internalization feature” . The internalization score is an important factor for fig 1C/ selection, but is not clear how it is done. Material and methods are just referring to Internalization feature, but what kind of algorithmic and criteria is behind this? 

The equation used is the following: 

IS=log⁡(a/(1-a)) where a=(mI/(mI+mB))*(pI/PB) where 

B = External and I = Internal part of the cells, mI = Mean intensity of upper quartile pixels in I, mB = Mean intensity of upper quartile pixels in B, pI = Peak intensity of upper quartile pixels in I, pB = Peak intensity of upper quartile pixels in B.

We have added it in the material and methods section to improve understanding (lines 330-337) rather than in the results section (we have removed lines 537). We have also defined the normalized internalization score as the calculated ratio of the internalization score measured at 37°C divided by the internalization score at 4°C. The normalized internalization score is now shown on modified Fig 1D and on Table 1.

Uptake in endothelia cells grown on slide or Transwells could be highly supportive. E.g in Fig 4D. Why the authors not co-stain against TfR and 4F2 with the Zo staining? Could give important information of cellular localization and thereby sorting mechanism.

We agree with the reviewer that the target of the 4F2 antibody would be very informative and could be explored with colocalization experiments as you suggested. We have not carried out colocalization experiments but we performed immunofluorescent cell labeling on hCMEC/D3 cells in red for 4F2, in blue for nuclei in the picture below. The 4F2 target is present on the cell membrane (see picture below):

We added the information line 733 and in supplementary data S8 Fig

 (Why is so many control antibodies transcytosed in the carton (Fig 2D)?) 

You are totally right; we have modified the picture with pink medium and less control antibodies transcytosed. You can see the new sketch below and we have modified the figure 2.

Tightness; ZO1 staining is nice, but is not a valid method to demonstrated tightness. Control antibody is included in most transcytosis experiments, but the result is give as ratios. The Papp would have been valuable for this judgement. And the TEER values is not supportive for the cells being tight. 

We agree that even though TEER values are strong indicators of the integrity of the cellular barriers before they are evaluated for transport of drugs or chemicals, they do not provide sufficient information on a restrictive paracellular pathway. That’s why we included in each well aside from the human IgG for which we determined Papp, a control mouse IgG to confirm paracellular integrity. In fact, control antibodies Papp are represented in figures 4A, 4B and 4C. This, along with ZO1 labeling supported that there is no paracellular increase after the cell treatment with anti-integrin antibodies. ZO1 labeling was performed to verify that the tight junctions were still present after treatment of the cells with anti-integrin antibodies. This experiment was inspired by Izawa Y, 2018 and Osada, 2011 reporting of a disruption of tight junctions after treatment of cells with anti-b1 antibodies. We have completed the Figure 4 with fluorescein Papp (Fig 4D) and TEER values (Fig 4E) measured in BrainPlotting model.

Normally there is a background in qPCR. Is there no background on e.g. GFAP in qPCR fig 1b? is the primers verified? 

We have used GFAP primers on cortex homogenates and we have detected GFAP after 21 PCR cycles amplification (21Ct) before Actin (24Ct). GFAP primers were verified. 

Line 438 “The results in Fig 1B show that even if the cells exhibit the right gene profile upon first isolation (P1D0) with high expression of CD31, CLDN 5 and BCRP and no..” what is the right gene profile? It is a few select genes know to be expressed in BEC, but many more (as well as polarity and efflux capacity) should be characterized to demonstrated it is a proper BEC. 

We wanted to verify the maintenance of the main BBB markers in BEC in culture. We observed that the main ones decrease so we can easily conclude on the loss of the endothelial phenotype even if other BBB markers could be maintained. We agree with you, the right gene profile is not precise we have replaced “the right gene profile” by “a brain endothelial gene profile” line 494.

It is an interesting approach to immunizes with BEC lysate; it does not find targets that is preferential expressed in brain endothelial cells. The data in fig 5 underscores that there might be a high distribution to other tissues. Do the authors think that this target would be relevant for clinical trials – discuss?

A brain-specific target would be the best candidate, we fully agree with the reviewer and for this reason our next campaign would include an adsorption of hybridoma supernatants on endothelial cells from peripheral tissues. However, there are presently no such brain-specific target and the ones that are currently used in development such as Transferrin, insulin or IGR1 receptor are ubiquitous but have proven useful to enhance brain exposure. For TfR in particular, one such compound is already on the market (fusion protein with Iduronate sulfatase approved for treatment of mucopolysaccharisose in Japan, Izcargo®) validating the concept and that safety is not a major concern. More particularly, for integrins, integrin alpha 7, also ubiquitous but with a high muscular expression, has been reported useful to target biologics to muscle Baik 2020

Finally, α3β1 and α5β1 integrins are attractive and tumor-selective pharmacological targets in cancer or metastases (Schaffner, 2013; Chen 2018; Kim, 2000; Milner, 2002; Labus, 2018), angiogenesis (Schaffner, 2013), in brain but also periphery. 

It was questioned if 4F2 binds to luminal and or abluminal membrane (p 822-24 Our confocal microscope immunofluorescence study (Fig 5D) did not allow to determine whether this labeling with 4F2 was luminal or abluminal). An easy experiment would be to use a Transwell NHP cells in a tight monolayer and the add the antibody respectively to each side and follow endocytosis with IF (e.g. 1 hour chase and fix and IF). This would answer the question which is highly relevant 

We agree with the reviewer. In fact, this is what we have done using the human brain primary model from Brainplotting. We measured the permeability ratios from apical to basolateral (influx) and from basolateral to apical (efflux). In the graph below we can see that the influx/efflux ratio is 1 suggesting that the target of 4F2 could be expressed as much on the apical side as on the basolateral side. In comparison the influx/ efflux ratio for an anti-TfR antibody was >2.

Reviewer #2: 

Summary

Overall, this manuscript describes a robust method of producing and identifying BBB targeting antibodies using a mouse immunized with NHP BMECs. This screen identified a highly enriched population of integrin binding antibodies which are suggested to undergo transcytosis. The authors use multiple different in vitro models of human, NHP, and immortalized human BBB models to verify the antibodies transcytosis ratio compared to an internal control. Comparing their lead candidates to integrin antibodies currently on the market or in development to identify possible methods of action. While they were unable to show the method of transcytosis, or if it is able to carry and deliver cargo across the endothelium, this work identified integrins as a potential target for BBB delivery for the first time. The work, while interesting, is lacking statistical support for the conclusions and confirmation that integrin binding is leading to transcytotic or at least vesicle based trafficking.

Major comments

Figure 5 is difficult to interpret as a whole. The figure legend is not descriptive enough to interpret the data being presented. For instance, adding information of the color indications to the caption would be beneficial. 

We thank the reviewer for the comment. We have changed the figure 5 along with the legend in lines 744 to 750. Images quality was improved, scale bars were indicated.

 Similarly, the complete lack of any scale bars or indications of the size of the field of view make it difficult to discern what the structures are that we are observing, 5D has scale bars, but the size of the scale bars is never indicated. The isotype column in 5C was never described, it’s unclear what tissue this is in and why the most important brain isotype controls are lacking in this figure. These should be included. 

We have added all isotype controls for the figure 5C in supplementary data S6 They are copied below for your convenience.

Statistical testing to support conclusions is completely absent. There needs to be description of statistical tests in the methods and confidence (p-values) given in the figure legends and/or figures. The form of replication and number of replicates also needs to be reported. 

We thank the reviewer for the recommendation, we have added statistical analysis for our internal models in figure 2 and we added the explanation in the materials and methods section lines 357-362 and lines 380-385 and in the legend of the figure 2 lines 573-576 and 578-581. We indicated line 563 data are available in the supplementary data S2 Table.

For each parameter, only antibodies with at least 3 measurements were included in statistical analysis. 

For pulse chase ratio analysis, which were meant to rank order the antibodies based on their ratio of basolateral vs apical concentrations, a margin of 0.1 was defined and a superiority analysis was conducted for each antibody using a one sample. In order to control multiplicity at 2.5% level, the p-values were adjusted using the Bonferroni-Holm correction.

For Papp ratio, a superiority analysis was conducted using an analysis of variance (ANOVA) to compare each antibody versus negative control followed by one-sided Dunnett’s test to control multiplicity at 2.5% level. Prior to statistical analysis, data were subjected to inverse transformation to ensure normality of residuals.

The analysis was performed using SAS 9.4 for Windows 10.

***: p-value <0.001

**: 0.001< p-value <0.01

*: 0.01< p-value <0.025

NS: p-value >0.025

We have replaced the figures in the original paper (Fig2) by the ones hereunder including these statistical analyses.

What is the method of transcytosis of these integrin antibodies given that integrins do not typically function in a vesicle trafficking modality? 

Integrins have been reported to perform endocytosis and recycling through several mechanisms, clathrin-dependent or -independent (caveolin-dependent, macropinocytosis, others). For a good review on the subject see Paulina Moreno-Layseca, Nat. Cell. Biol. 2019. Transcytosis on the other hand has been much seldomly reported and mostly on epithelial cells (Ivanenkov, 2000; Staquicini 2021; Su 2017; Xu 2016; Kassa 2019; Alfsen 2005 Very few papers discuss transcytosis across brain endothelial monolayers mediated by integrin (mainly β1 subtype). In all cases, the objects for which transcytosis was observed were particles (nanoparticles, cells, …) either expressing or decorated with integrin ligands (Magnussen 2020; Malin 2014; Su 2017, Ruan 2017, Wei 2015)). In fact, after submitting our paper we came across a very relevant article from Deshayes 2021 demonstrating that Streptococcus group B are using α5β1 and αvβ3 for brain invasion in juvenile meningitis. We have added into our discussion lines 835-837. Nevertheless, no anti integrin antibody has been shown to be able to transcytose cells. 

In the discussion the authors discuss the use of integrin β1 blocking antibodies which would be helpful data to include to justify the conclusions. 

We totally agree. A few authors have shown that blocking anti integrin β1 antibodies were able to prevent uptake or transcytosis of the particles referred above ( Su 2017, Deshayes 2021). We could have performed our transcytosis experiments in the presence of some of these blocking antibodies. Alternatively, we had planned to perform transcytosis with endothelial cells knocked down for some specific integrin subunits with siRNA or CRISPR. 

High resolution imaging would also be useful to show co-localization of the integrin and targeting antibody within cellular endosomal/lysosomal/etc structures. 

We totally agree. We have performed fluorescent IHC in NHP brain capillaries from frozen healthy NHP brain slices (Fig. 5D of the article). The images show a nice co-localization of the 4F2 anti integrin mAb with the endothelial cells. However, higher resolution to distinguish between luminal and abluminal side turned out not exploitable. Co-staining with a luminal PgP marker in a higher resolution setting could answer the question. On the other hand, using the human brain primary model from Brainplotting, we measured the permeability ratios from apical to basolateral (influx) and from basolateral to apical (efflux). In the graph below we can see that the influx/efflux ratio is 1 suggesting that the target of 4F2 could be expressed as much on the apical side as on the basolateral side. In comparison the influx/efflux ratio for an anti TfR antibody was >2.

One of the downfalls of this method seems to be the lack of murine cross-reactive antibodies. As these were raised in an immunized mouse model this is not terribly surprising. The authors should discuss how the platform presented can 1. Lead to antibodies that are difficult to test preclinically and 2. Not lead to antibodies against conserved targets between mouse and NHP. 

Here also we fully agree with the reviewer. The chance to obtain mouse-human or mouse-NHP cross-reactive antibodies for highly conserved target between mouse and human (or NHP) from an alternative species immunization (Lama to generate nanobodies, humanized chicken (as reported by Kathryn H. Ching) would have been higher if the target is not highly conserved between mouse and lama or mouse and chicken. Nevertheless, mouse-human cross-reactive antibodies have been reported through mice immunization (we have an example internally with an anti FGFR4 antibody WO2010/004204 (2010) and some examples in the literature : anti-PD1 Huang 2010, anti-claudin 4 Hashimoto 2016, anti-VEGFR2 Li 2021. The primary goal of our approach with this platform was to screen antibodies with internalization and transcytosis capabilities from an immunization strategy with primary non-human primate brain microvascular endothelial cells. In case no cross-reactive antibody would be identified, our plan was to either generate antibodies against the identified promising targets through library screening using recombinant proteins or cells expressing the antigen, or to perform affinity maturation of our identified antibodies to obtain binders to the mouse and human derived targets. These possibilities were discussed in the discussion section line 961: “Inherently to the host, it is notoriously difficult to obtain cross reactive antibodies for highly conserved mouse-human targets during mouse immunization campaigns. Alternative solutions would be to screen naïve antibody libraries against human and mouse brain primary endothelial cells to select cross reactive clones or to immunize lama and select cross reactive nanobodies” and later line 971: “From this structure, the potential to perform affinity maturation to get mouse-human cross-reactive antibodies could be assessed”.

As the authors point out these antibodies seem to be targeting pan-endothelial markers, and one of the problems mentioned in the introduction is the lack of brain specific targeting therapeutics.

A brain-specific target would be the best candidate, we fully agree with the reviewer and for this reason our next campaign would include a counter screen for peripheral tissues. However, there are presently no such brain-specific target and the ones that are currently used in development such as Transferrin, insulin or IGR1 receptor are ubiquitous but have proven useful to enhance brain exposure. For TfR in particular, one such compound is already on the market (fusion with Iduronate sulfatase approved for treatment of mucopolysaccharisose in Japan, Izcargo®) validating the concept and that safety is not a major concern. 

 Could the authors comment on if there is anything known about brain specific or brain enriched integrin heterodimers that could be investigated for this purpose? If as expected, there is not evidence of brain integrin isoform enrichment, the authors should at least speak to the impact of targeting a pan-endothelial protein for CNS drug delivery. 

We performed analyses of all α and β monomeric subunits and their dimeric combinations in gene and protein expression databases and even if several of them are highly present in the brain their abundance is not higher than in some other tissues. Nevertheless, in addition to our response above regarding ubiquitous targets for brain delivery, integrin alpha 7, also ubiquitous and which expression profile does not substantially differ from the other integrins has been reported useful to target biologics to muscle (Andrew D. Baik,, 2020).

Minor comments

The human primary BMEC data included in figure 1A and B should be removed as these cells are not used in any subsequent panel in the paper. 

During the course of our work, it was a challenge to generate human primary endothelial cells of good quality and in number. In fact, we were unsuccessful with all attempts and strategies we undertook, even from relatively fresh postmortem human brain, or through subcontractors. We also analyzed external sources of human primary brain cells and found some of them of very poor quality and for the ones that passed quality control much too expensive to envisage purchasing the high amounts we needed for our immunization campaigns. We were surprised that these difficulties are seldomly mentioned in the literature and thought our experience could be useful to the man of art. We have included these data into the supplementary figure S1. 

Instead, it would be more beneficial to include similar validation data for the NHP BMECs that were actually used for immunization and include these data in the supplement or new Figs 1A and 1B, even if this data is essentially all represented in the Chaves et al paper. E.g. characterization of the cells actually used to immunize in this paper. 

We thank the reviewer for these remarks, we have changed the Figure 1 A, B and C with selected data from Chaves et al paper. Indeed, we have used the NHP BMEC characterized in this paper to perform the mice immunization. We have added the cells functionality in Figure 1C. We have commented the new figure 1 from lines 523 to 528 and we have added protocols in material and methods section: 

Immunofluorescence experiments from lines to 234-240

RNA seq experiments from lines to 241-263

Transcytosis experiment from lines 365 and 367

While the authors affirm that the NHP model has good barrier properties, it would be extremely beneficial to describe some of the important characteristics of this model. Particularly as in line 491 the NHP cells were described as not having strong tight junctions and justified the use PCR, but in line 519 these are apparently tight enough to do a traditional transcytosis assay. 

The features and characteristics of the NHP BBB model that was used are included in the paper Fig1 and lines 523-528. Nevertheless, inherently to primary cultures, their quality can vary between different productions and animals. Therefore, we systematically evaluated the TEER of our cultures. If the TEER was above our selected criteria (150 Ω.cm2) we engaged them in our transcytosis model. Nevertheless, we always included an internal control in each well (a mouse irrelevant IgG) to ensure paracellular permeability was tight. When the control was above the expected value, the experiment was invalidated. If the TEER value was under our threshold of 150 Ω.cm2 , the Transwell were used in a pulse chase protocol (similar to the one reported by Sade, 2014)

Figure 1A needs to have a scale bar indicating the size of the cells. 

OK we have added the magnification in the figure caption in supplementary data S1 Fig. Thanks.

Figure 1C contains a red squiggle indicative of a misspelled word on the y-axis. This graph would also benefit from pointing out the data points that correspond to the images selected. We thank the reviewer and we have corrected the figure 1 D

How were the internalization score bins determined? 

The equation used is the following: 

IS=log⁡(a/(1-a)) where a=(mI/(mI+mB))*(pI/PB) where 

B = External and I = Internal part of the cells, mI = Mean intensity of upper quartile pixels in I, mB = Mean intensity of upper quartile pixels in B, pI = Peak intensity of upper quartile pixels in I, pB = Peak intensity of upper quartile pixels in B.

We have included the equation that was used to determine the internalization scores along with the definition of the parameters in the materials and methods sections lines 330-337. We have also defined how the normalized internalization score was calculated on the x axis in Fig 1D and in Table 1 to facilitate the bin classification reported on lines 538.

And why are absolute internalization scores not reported in table 1 rather than the low/high bins? 

The normalized internal score is now reported on Table 1.

Line 644 has an unpaired parenthesis around Figure 4A-C. OK we have corrected. Thanks

How does TJ morphology and ZO-1 localization and expression compare to before incubation with antibodies? (Fig 4D). 

We agree it would be interesting but in fact the images after the experiments have been obtained after fixing the cells. This treatment would not be compatible with the experiment a posteriori. 

The authors should include the raw Papp data for the control antibodies as described in line 633-634. This is very important to exclude integrin binding as disruptive. 

Figure 4A, B and C are in fact showing the results of all Papp data for the control antibodies in the 3 models (non-human primate internal model, Pharmacocell, BrainPlotting models) lines 717 and 718. We have reported mouse control IgG Papp in cm.min-1 for each tested antibody. This is one of the indications that integrin binding has not disrupted the barrier integrity. For the BrainPlotting model we have completed the figure 4 with fluoresceine control Figure 4D and TEER measurements Figure 4E.

Lines 87-90 speak to the novelty and importance of immunized libraries. The authors should cite a very recent paper from Lajoie et al. scientific reports, 2022 (https://doi.org/10.1038/s41598-022-09962-8) where an analogous strategy was deployed for CNS targeting purposes.

Absolutely. We were aware of the work of Shusta in this area as we had seen his proposal. Their publication came out 3 days before we submitted our paper. Therefore, we added the information from lines 94 to 97 and the reference.

---

## [Decision Letter · Decision Letter 1]

9 Aug 2022

PONE-D-22-11645R1An innovative strategy to identify new targets for delivering antibodies to the brain has led to the exploration of the integrin familyPLOS ONE

Dear Dr. Cegarra,

Thank you for submitting your manuscript to PLOS ONE. After careful consideration, we feel that it has merit but does not fully meet PLOS ONE’s publication criteria as it currently stands. Therefore, we invite you to submit a revised version of the manuscript that addresses all the points raised during the review process.

The manuscript has been improved, only some minor issues remained to be corrected as specified by Reviewer 2.

We look forward to receiving your revised manuscript.

Kind regards,

Mária A. Deli, M.D., Ph.D.

Academic Editor

PLOS ONE

Journal Requirements:

Reviewers' comments:

Reviewer's Responses to Questions

**Comments to the Author**

1. If the authors have adequately addressed your comments raised in a previous round of review and you feel that this manuscript is now acceptable for publication, you may indicate that here to bypass the “Comments to the Author” section, enter your conflict of interest statement in the “Confidential to Editor” section, and submit your "Accept" recommendation.

Reviewer #1: All comments have been addressed

Reviewer #2: (No Response)

2. Is the manuscript technically sound, and do the data support the conclusions?

Reviewer #1: Yes

Reviewer #2: Yes

3. Has the statistical analysis been performed appropriately and rigorously? 

Reviewer #1: Yes

Reviewer #2: Yes

4. Have the authors made all data underlying the findings in their manuscript fully available?

Reviewer #1: Yes

Reviewer #2: Yes

5. Is the manuscript presented in an intelligible fashion and written in standard English?

Reviewer #1: Yes

Reviewer #2: Yes

6. Review Comments to the Author

Reviewer #1: All questions have been properly answered. The is no further comments.

Reviewer #2: The authors responded well to all of the comments and suggestions made. This manuscript represents an extremely thorough attempt to screen for an identify antibodies for BBB targeting and transcytosis. The combination of in vitro assays was impressive and is as convincing as can be reasonable expected. The remaining comments are minor and are related to some polishing and clarification.

The pulse chase assay was performed with cells with TEER below 150 ohm cm2, but the transcytosis assay was described as using cells with a high TEER but is also described as having at TEER up to 150 ohm cm2. Was there a minimum TEER value considered in either situation? Was there a difference in the NHP cells selected for the two different techniques? This should be explicitly state in either the results narrative or the materials methods.

Citation(s) is needed on line 65 outlining the additional identified receptors.

The paragraph lines 71-81 is almost directly duplicated in the next paragraph. All of this is good information to back up and ground this manuscript, but it would benefit from some polishing.

Figure 2 – it would improve readability if the bars were ordered the same in C as they are in B and E

7. PLOS authors have the option to publish the peer review history of their article (what does this mean?). If published, this will include your full peer review and any attached files.

Reviewer #1: No

Reviewer #2: No

---

## [Author Response · Author response to Decision Letter 1]

11 Aug 2022

Reviewer #2: The authors responded well to all of the comments and suggestions made. This manuscript represents an extremely thorough attempt to screen for an identify antibodies for BBB targeting and transcytosis. The combination of in vitro assays was impressive and is as convincing as can be reasonable expected. The remaining comments are minor and are related to some polishing and clarification.

The pulse chase assay was performed with cells with TEER below 150 ohm cm2, but the transcytosis assay was described as using cells with a high TEER but is also described as having at TEER up to 150 ohm cm2. Was there a minimum TEER value considered in either situation? Was there a difference in the NHP cells selected for the two different techniques? This should be explicitly state in either the results narrative or the materials methods.

In fact, it was an error, so line 367, we have replaced “up to 150 Ω.cm2” by “from 150 Ω.cm2”. 

Citation(s) is needed on line 65 outlining the additional identified receptors.

The paragraph lines 71-81 is almost directly duplicated in the next paragraph. All of this is good information to back up and ground this manuscript, but it would benefit from some polishing.

To address the two comments above we have reworked the introduction lines 70,71, 80,81 and 84.

Figure 2 – it would improve readability if the bars were ordered the same in C as they are in B and E

We thank the reviewer for this observation, and we have corrected the Figure 2.

---

## [Decision Letter · Decision Letter 2]

2 Sep 2022

An innovative strategy to identify new targets for delivering antibodies to the brain has led to the exploration of the integrin family

PONE-D-22-11645R2

Dear Dr. Cegarra,

We’re pleased to inform you that your manuscript has been judged scientifically suitable for publication and will be formally accepted for publication once it meets all outstanding technical requirements.

Kind regards,

Mária A. Deli, M.D., Ph.D.

Academic Editor

PLOS ONE

Additional Editor Comments (optional):

Reviewers' comments:

Reviewer's Responses to Questions

**Comments to the Author**

1. If the authors have adequately addressed your comments raised in a previous round of review and you feel that this manuscript is now acceptable for publication, you may indicate that here to bypass the “Comments to the Author” section, enter your conflict of interest statement in the “Confidential to Editor” section, and submit your "Accept" recommendation.

Reviewer #2: (No Response)

2. Is the manuscript technically sound, and do the data support the conclusions?

Reviewer #2: Yes

3. Has the statistical analysis been performed appropriately and rigorously? 

Reviewer #2: Yes

4. Have the authors made all data underlying the findings in their manuscript fully available?

Reviewer #2: Yes

5. Is the manuscript presented in an intelligible fashion and written in standard English?

Reviewer #2: Yes

6. Review Comments to the Author

Reviewer #2: (No Response)

7. PLOS authors have the option to publish the peer review history of their article (what does this mean?). If published, this will include your full peer review and any attached files.

Reviewer #2: No

---

## [Editor Report · Acceptance letter]

6 Sep 2022

PONE-D-22-11645R2 

An innovative strategy to identify new targets for delivering antibodies to the brain has led to the exploration of the integrin family 

Dear Dr. Cegarra:

I'm pleased to inform you that your manuscript has been deemed suitable for publication in PLOS ONE. Congratulations! Your manuscript is now with our production department. 

Kind regards, 

on behalf of

Prof. Mária A. Deli 

Academic Editor

PLOS ONE